# Unlocking the mysterious polytypic features within vaterite CaCO$_3$

Xingyuan San [1,6], Junwei Hu [2,6], Mingyi Chen[2], Haiyang Niu [2]✉,
Paul J. M. Smeets [3], Christos D. Malliakas [4], Jie Deng[5], Kunmo Koo [3],
Roberto dos Reis [3], Vinayak P. Dravid [3]✉ & Xiaobing Hu [3]✉

Calcium carbonate (CaCO$_3$), the most abundant biogenic mineral on earth, plays a crucial role in various fields such as hydrosphere, biosphere, and climate regulation. Of the four polymorphs, calcite, aragonite, vaterite, and amorphous CaCO$_3$, vaterite is the most enigmatic one due to an ongoing debate regarding its structure that has persisted for nearly a century. In this work, based on systematic transmission electron microscopy characterizations, crystallographic analysis and machine learning aided molecular dynamics simulations with ab initio accuracy, we reveal that vaterite can be regarded as a polytypic structure. The basic phase has a monoclinic lattice possessing pseudohexagonal symmetry. Direct imaging and atomic-scale simulations provide evidence that a single grain of vaterite can contain three orientation variants. Additionally, we find that vaterite undergoes a second-order phase transition with a critical point of ~190 $K$. These atomic scale insights provide a comprehensive understanding of the structure of vaterite and offer advanced perspectives on the biomineralization process of calcium carbonate.

Calcium carbonate (CaCO$_3$) is one of the most abundant natural materials found in biomineral systems, which plays a pivotal role in the chemistry of the hydrosphere, lithosphere, biosphere, climate regulation, and scale formation[1–4]. Among the three anhydrous crystalline structures, vaterite[5], aragonite[6] and calcite[7], vaterite has the least stability. It is frequently observed as a transient intermediate phase during the transition process of amorphous calcium carbonate to other more stable polymorphs[8,9] or formed via liquid–liquid phase separation in supersaturated CaCO$_3$ solution[10–12], thereby kinetically directing the mineralization pathway. Although vaterite is less stable and rarer than other forms of CaCO$_3$, it can still be commonly found in many biogenic and abiotic systems, such as freshwater lackluster pearls[13], cement[14] and natural minerals[15]. In addition, vaterite's unique

structure, high solubility in water, and porosity properties relative to other crystalline polymorphs make it an ideal functional material for environmental chemistry[16,17], bone tissue[18,19] and biomedical engineering[5,20]. Therefore, an accurate description of the structure of vaterite at the atomic scale is useful for understanding the growth dynamics of calcium carbonate[10,21–23] and fulfilling its potential applications as a functional material[24–26].

Despite decades of experimental and theoretical efforts[27–31], the precise structure of vaterite remains a subject of debate (See Supplementary Note I and Supplementary Table 1). Several conflicting structures have been proposed, such as the hexagonal or rhombohedral structures[29,32,33], orthorhombic structure[34], monoclinic lattice and triclinic lattice[29,30]. The discrepancies arise not only from the

[1]Hebei Key Lab of Optic-electronic Information and Materials, The College of Physics Science and Technology, Hebei University, Baoding 071002, China. [2]State Key Laboratory of Solidification Processing, International Center for Materials Discovery, School of Materials Science and Engineering, Northwestern Polytechnical University, Xi'an 710072, China. [3]Department of Materials Science and Engineering, The NU*ANCE* Center, Northwestern University, Evanston, IL 60208, USA. [4]Department of Chemistry, Northwestern University, Evanston, IL 60208, USA. [5]Department of Geosciences, Princeton University, Princeton, NJ 08544, USA. [6]These authors contributed equally: Xingyuan San, Junwei Hu. ✉e-mail: haiyang.niu@nwpu.edu.cn; v-dravid@northwestern.edu; xbhu@northwestern.edu

ordering of carbonate groups, particularly the slight rotation within a single layer, but also from the long-range stacking sequence of carbonate layers along the z-axis concerning a hexagonal lattice. Recently, by means of aberration-corrected high-resolution transmission electron microscopy (HRTEM), Pokroy et al revealed that vaterite crystals contain two interspersed crystal structures, namely a predominant hexagonal Kamhi lattice together with an unknown structure. Although a combined Kamhi and Meyer model can be used to explain the strong diffraction spots, neither one can explain any of the weak diffractions[31]. Thus, the actual fine details of the vaterite structure remain a mystery. Until now, it has been generally acknowledged that vaterite cannot be described using a single long-range ordered structure, and it likely shows considerable micro-twinning and stacking disorder[27,34,35]. However, to our knowledge, direct imaging evidence of various types of stacking disorders at the atomic scale in vaterite is still missing.

Uncovering the structural features of vaterite through experimental means is a very challenging task. Conventional bulk measurements such as X-ray diffraction (XRD), electron diffraction and Raman spectra are not sufficient, and atomic resolution imaging is a prerequisite. HRTEM imaging is one choice, however, this technique is actually very limited in resolving the complex structure here since it can only provide a phase contrast image that usually contains many artifacts caused by thickness and focus variations. In contrast, atomic resolution high angle annular dark field (HAADF) imaging is an ideal technique since it provides the atomic number (Z) contrast[36], with the intensity approximately in proportion to $Z^2$. Here, we uncover the structural mystery of vaterite by combining atomic resolution experiments and molecular dynamics simulations adopting a deep neural network (DNN) interatomic potential with ab initio accuracy. As vaterite is a metastable crystalline form of $CaCO_3$, its crystallinity, no matter synthesized or naturally existed, is usually not good. Here we have conducted our experimental analysis based on samples from two different sources, the lackluster pearls and the asteriscus of carp. The lackluster pearl is a representative sample having good crystallinity, which is very suitable for high quality atomic resolution imaging analysis. To verify that our extracted information is appliable to other vaterite samples, we performed detailed observations on another vaterite sample with relatively good crystallinity, namely the asteriscus otolith pairs. The results of the two paradigms are complementary and can be cross validated. Our atomic scale insights into the vaterite structure resolve the contentions that existed among previous research work and provide an intrinsic perspective for studying the biomineralization mechanism of $CaCO_3$, which may consequently facilitate the synthesis of calcium carbonate based materials with target structure and performance.

## Results

### General structural features of vaterite

We first measured the Raman spectra of the lackluster pearls. As shown in Fig. 1a and Supplementary Fig. 1a, the splitting Raman band $v_1$, which represents the symmetric stretching mode of the carbonate ions, has the highest intensity. The spectral features shown here are a good match to those obtained from geological, synthetic and other biogenic samples comprising vaterite minerals[37-39], indicating that our lackluster sample can be categorized as a common biomineral containing a vaterite structure. The most intense peak is at 1091 cm⁻¹, with consecutively lower intensities for the peaks at 1075 cm⁻¹ and 1080 cm⁻¹. The triple splitting suggests the presence of at least three crystallographically independent carbonate groups and similar carbonate group layers within vaterite[40,41]. In addition, as shown in Supplementary Fig. 1b, our XRD data agree well with other diffraction data of vaterite in the literature[37-39]. This further confirms that our lackluster pearls indeed consist of representative biominerals with a vaterite structure.

Figure 1b presents a HAADF image of the lackluster pearl, which shows a laminar structure at the micrometer scale. The associated elemental maps, shown in Supplementary Fig. 1c–e, confirm the uniform distribution of C, Ca and O in the grain interior. Additionally, the HRTEM image in Fig. 1c reveals the general stacking feature of vaterite at the atomic scale. The inset digital fast Fourier transformed (FFT) patterns suggest the presence of many disordered stacking layers along the $(001)_H$ plane, labeled based on the hexagonal Meyer lattice[32]. This finding is consistent with previous results on other vaterite samples from different originations[30,31,35]. However, due to the significant offset resulting from the phase contrast, the HRTEM image in Fig. 1c cannot be used to extract further structural details.

To uncover the intrinsic structural features of vaterite, we then performed large angle tilting experiments to obtain serial selected area electron diffraction (SAED) patterns from a single grain. The relative positions and associated experimental tilting angles of each diffraction pattern in reciprocal space are given schematically in Supplementary Fig. 2, and diffraction patterns are displayed in Fig. 2. The diffraction patterns in Fig. 2a reveal a hexagonal symmetry feature, which is further confirmed by convergent beam electron diffraction and atomic resolution imaging analysis (See Supplementary Fig. 3). Such results agree well with the suggested hexagonal structure[42,43]. However, the diffraction patterns in Fig. 2b–f can only be indexed by a monoclinic structure with lattice parameters of $a = 12.2$ Å, $b = 7.2$ Å, $c = 9.3$ Å and $\beta = 115.2°$. These lattice parameters could match any of three monoclinic lattices including the C2, Cc, and C2/c structures proposed in previous work[29]. Unfortunately, diffraction data (See Supplementary

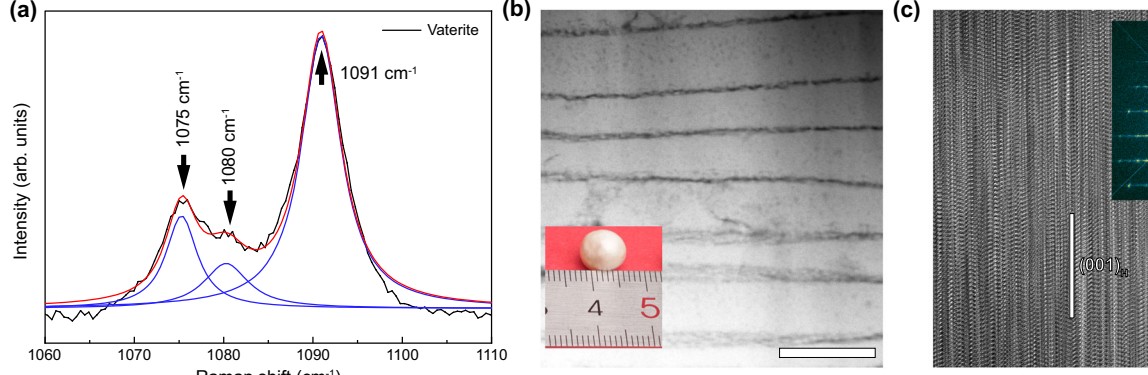

**Fig. 1 | Microstructural features of vaterite. a** Splitting of the most intense Raman band $v_1$. The best result of the Gaussian fitting procedure was achieved for the decomposition of three peaks at 1075 cm⁻¹, 1080 cm⁻¹ and 1091 cm⁻¹. **b** HAADF image showing the laminate structure. The inset shows the morphological features of the pearls with vaterite structure. Scale bar = 500 nm. **c** HRTEM image of vaterite. Inset is the corresponding FFT pattern. Streaking features in inset FFT patterns indicate the representative stacking feature in vaterite structure. Scar bar = 10 nm.

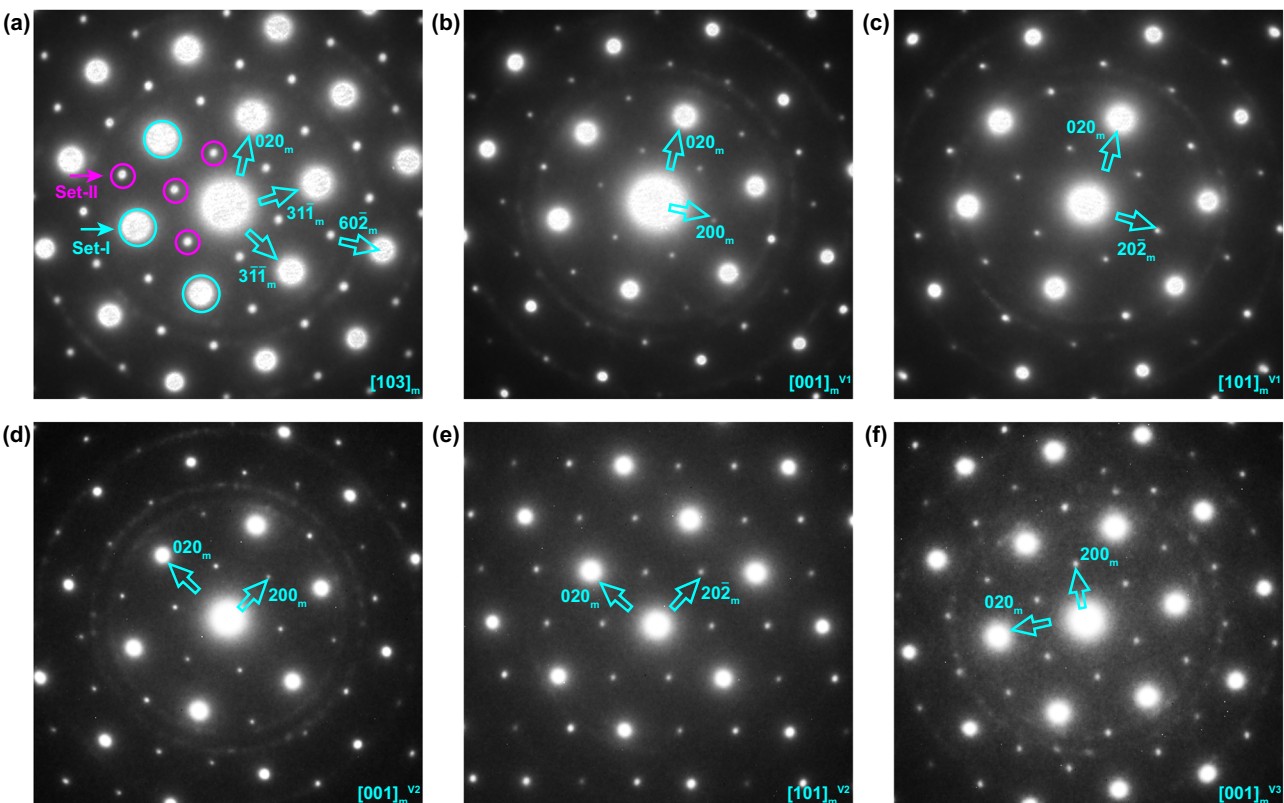

**Fig. 2 | A series of SAED patterns of vaterite obtained from a single grain.** EDPs along the (**a**) $[103]_m$, (**b**) $[001]_m^{V1}$, (**c**) $[101]_m^{V1}$, (**d**) $[001]_m^{V2}$, (**e**) $[101]_m^{V2}$ and (**f**) $[001]_m^{V3}$ directions. The subscript $m$ represents the monoclinic lattice. The superscripts $V1$, $V2$ and $V3$ indicate three orientational variants. Circles in **a** indicate two sets of reflections with significant differences in intensities with Set I/II patterns having higher/lower intensities.

Figs. 4, 5 and Note III) cannot distinguish between the different rotations of carbonates of these three structures. The structural projection of the *C2* structure along the [010] direction is illustrated in Supplementary Fig. 8a, which can also be described using a hexagonal lattice as highlighted by the dashed line. In fact, the aforementioned monoclinic lattice is closely related to the disordered Meyer structure with a hexagonal lattice ($P6_3/mmc$; $a = b = 7.169$ Å, $c = 16.98$ Å). Their intrinsic relationships are discussed in Supplementary Note IV. As shown in Supplementary Fig. 8b, calcium ions ($Ca^{2+}$) approximately form a hexagonal array, and carbonate ions ($CO_3^{2-}$) occupy half of the $Ca_6$ trigonal prismatic interstices in one layer and the other half in the neighboring layers. In each interstice, the carbonate has three possible orientations, and the carbonate ions within a plane are regularly distributed resulting in an ordered arrangement that is more energetically favored[44]. Considering the rotational invariant, the stacking sequence composed of two adjacent layers is always identical (See Supplementary Fig. 9). Once a third layer is introduced, three possible stacking sequences can be obtained, *i.e.*, the carbonate groups in the third layer with three possible orientations correspond to rotations of +120°, −120°, or 0° around the *z*-axis compared to the counterparts in the first layer (See Supplementary Fig. 10). The three stacking sequences can be marked as "+", "−", and "0", respectively, following the notation originally proposed by Christy[28] (See Supplementary Note II and Supplementary Fig. 8c). The *C2* phase shown in Supplementary Fig. 8a and its two analogous structures, namely the *Cc* and *C2/c* structures share the same stacking sequence (+−+−+−), only the rotation of carbonates is slightly different.

In Fig. 2a, the electron diffraction patterns (EDPs) show two sets of reflections with different intensities, as indicated by circles. Only the spots with brighter contrast (Set-I patterns) match the monoclinic structure along the $[103]_m$ direction. In contrast, all of the reflections in Fig. 2b–f can be solely indexed along either the $[001]_m$ or $[101]_m$ direction based on the monoclinic lattice, despite the obvious intensity difference. Importantly, the relative distributions of the reflections in Fig. 2b, d, f are the same, except for a 60° rotation along the viewing direction. This phenomenon is also observed in Fig. 2c, e, indicating that there are likely three orientation variants of the monoclinic lattice within the vaterite structure (See Supplementary Fig. 11 and Supplementary Note V). These three variants can be formed by a 60° rotation along the $[103]_m$ zone-axis, considering the pseudohexagonal features of the monoclinic lattice. The stacking of variant segments along the $[103]_m$ direction yields the stacking disorder observed in Fig. 1c and the weaker spots in Fig. 2a–f (See Supplementary Note VI). Therefore, the domain concept proposed here can well rationalize the broadly existed weak spots that could not be explained in previous work[31]. Direct evidence of orientation variants in real space will be discussed below.

## Localized structural intergrowth within vaterite

To visualize the potential stacking order and/or disorder within vaterite, it is necessary to tilt the grain to the $[010]_m$ zone-axis, as this direction allows for edged-on projection of the planar defects. To facilitate experimental data analysis, we would like to uncover the slight difference along some $[010]_m$ related directions first. As demonstrated in Supplementary Fig. 11 schematic, the monoclinic lattice has a pseudohexagonal feature with a pseudo-6-fold axis along the $[103]_m$ direction. The $[010]_m$ direction is equivalent to the $[110]_m$ and $[1\bar{1}0]_m$ directions. Simulated EDPs along the above three directions are shown in Supplementary Fig. 14a–c, and structural projections of the above three directions are shown in Fig. 3a–c. It is observed that the stacking features of the Ca layers are nearly identical along the above three directions. In between two neighboring Ca layers along the vertical direction, there are layers composed of carbonates.

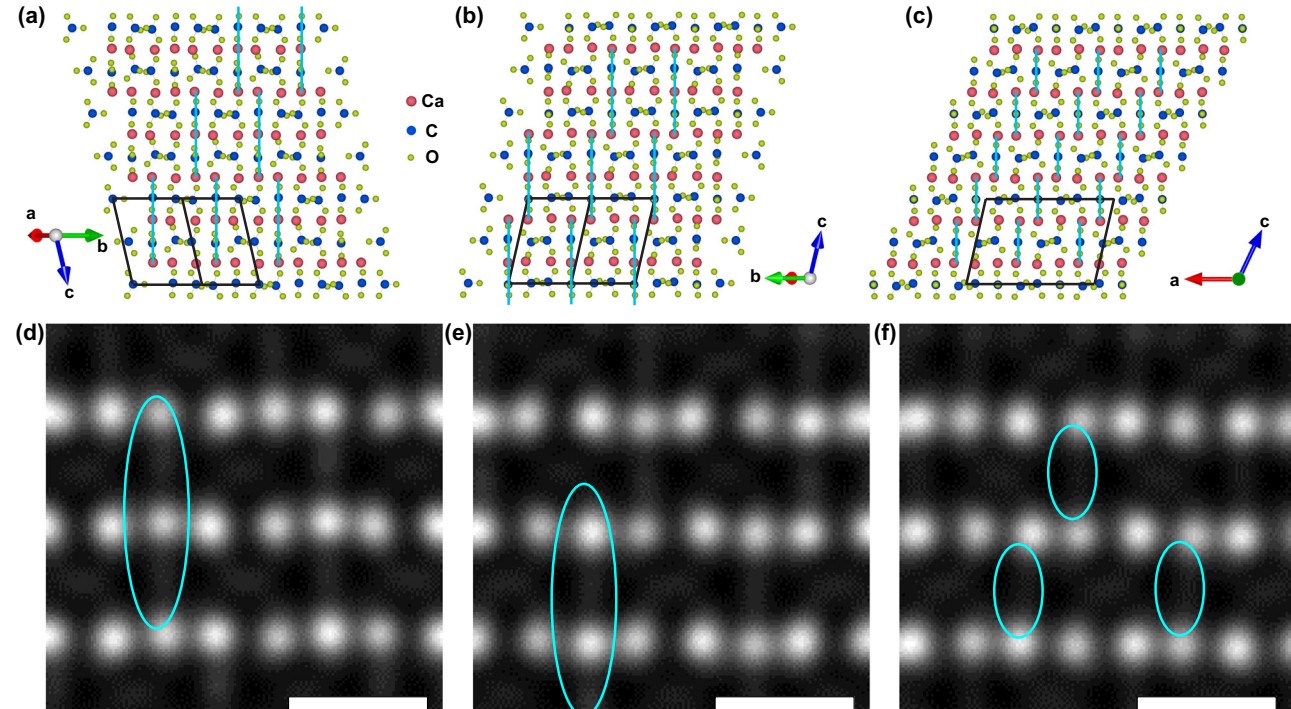

**Fig. 3 | Similarity and nuance difference for the stacking of carbonates along some directions.** Structural projection of the ordered *C2* vaterite along the (**a**) $[110]_m$, (**b**) $[1\bar{1}0]_m$, and (**c**) $[010]_m$ directions. The black frames indicate the projection of a unit cell along the corresponding directions. The cyan vertical lines indicate the stacking feature of the Ca-C-O chains along the $[001]_m$ direction. Arrows indicate the projected axes. The pink, blue and chartreuse balls indicate the Ca, C and O atoms respectively. Atomic resolution HAADF image simulations along the (**d**) $[110]_m$, (**e**) $[1\bar{1}0]_m$, and (**f**) $[010]_m$ directions. The cyan ovals in **d**–**f** indicate the diffused streaks along the vertical direction. Scale bar in **d**–**f** is 0.5 nm.

However, along these projected directions, the stacking features of the carbonate differ (indicated by vertical lines). Based on the atomic resolution HAADF image simulations shown in Fig. 3d–f, it is found that the C-O bonds orientated approximately along the vertical directions can generate the diffuse weak streak features indicated by cyan circles. Therefore, the stacking features of the diffuse streaks can be used to differentiate the stacking features of the carbonate layers.

In Fig. 4a, the experimental EDPs of vaterite are shown along the $[010]_m$ zone-axis, indicating a significant amount of stacking disorder on the $(001)_m$ plane due to the strong streaking in diffraction normal to this plane. Fig. 4b displays an atomic resolution HAADF image along this direction, while the corresponding strain map derived from the geometrical phase analysis is shown in Fig. 4c. The interface separating structures of different lattices are apparent, with the left and right regions labeled as A and B, respectively. Region A contains the stacking feature of monoclinic lattice along $[010]_m$ direction. However, the inset FFT patterns which corresponds to regain B can be indexed solely based on the monoclinic lattice along the $[110]_m$ direction. Within the vaterite grain interior, there are three variants, with variants II and III corresponding to the $[110]_m$ and $[1\bar{1}0]_m$ directions, respectively, if variant I is tilted to the $[010]_m$ direction. While structural projections along these three directions are very similar, as shown in Fig. 3, slightly different stacking sequences of carbonate layers, located between the two calcium layers, can be used to differentiate the variations in the local structural segment. Fig. 4b shows that some regions correspond to the monoclinic structure projected along the $[010]_m$ direction, while others correspond to the structural projection of the monoclinic lattice along the $[110]_m$ direction. The atomic resolution HAADF image with a larger field of view, shown in Supplementary Fig. 15, clearly demonstrates that the vaterite structure should be regarded as a polytypic structure with three orientation variants formed based on the monoclinic lattice.

It is worth noting that the atomic scale structural features obtained from lackluster pearls are reproducible in other vaterite samples of different originations. Specifically, we have compared characterization results with another vaterite biomineral, an asteriscus present in the otolith of carp, for further analysis. As demonstrated in Supplementary Fig. 16, the Raman spectrum of the asteriscus shows typical features of vaterite, and the HRTEM images and associated FFT patterns demonstrate the heavily faulted feature of vaterite grains, similar to what we found in the lackluster sample. Additionally, the atomic resolution HAADF image in Supplementary Fig. 16d shows the localized stacking features of calcium and carbonate layers, which can also be labeled based on the signs introduced in Supplementary Fig. 8. This direct evidence confirms that the primary structural information obtained from the lackluster pearl is the intrinsic information of vaterite and is generally applicable to other minerals containing a vaterite structure.

To validate our experimental results, we conducted molecular dynamics (MD) simulations to study the crystallization process of vaterite directly from liquid. We trained a DNN potential under the framework of the DeePMD approach[45], which can simulate a large system with ab initio accuracy for a long time[46–48], which is essential for crystallization simulation. To overcome the nucleation barrier, solid–liquid coexistence simulations (See Supplementary Fig. 18) were performed, starting with a single fixed Ca solid layer at $1000\,K$. The crystallization proceeded with the reorientation of the carbonate groups, and once completed, the system was quenched to $10\,K$ to eliminate thermal fluctuations. One of the final configurations is shown in Fig. 4d. Our simulation results reveal that the stacking sequence "0" is absent, while "+" and "−" are randomly distributed along the z-axis, equivalent to the $[103]_m$ direction in the monoclinic lattice (normal direction of the $(001)_m$ plane). These results are in excellent agreement with our experimental findings shown in Fig. 4b, Supplementary

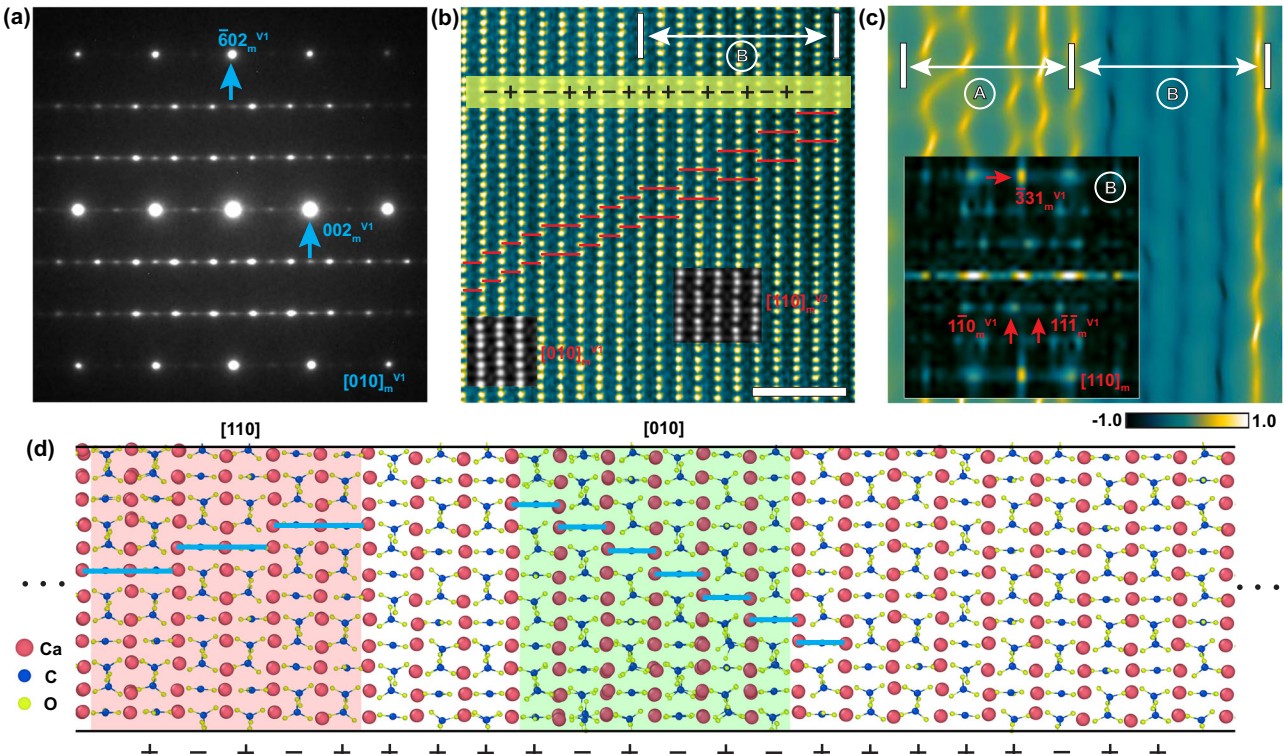

**Fig. 4 | Stacking disorder within vaterite. a** EDPs obtained from vaterite along the $[010]_m$ zone-axis. **b** Atomic resolution HAADF image along the $[010]_m$ zone axis showing the polytypic features within vaterite. The cyan horizontal lines indicate the stacking feature of the Ca-C-O chains along the $[001]_m$ direction. Inset image simulations correspond to the vaterite structure with the *C2* space group along the $[110]_m$ and $[010]_m$ directions. Scale bar = 2 nm. **c** Strain map ($\varepsilon_{xy}$) of the image shown in **b** obtained by geometric phase analysis. Inset FFT patterns corresponding to areas A. **d** Theoretically grown vaterite with an intergrowth of different polytypic structures. The pink, blue and chartreuse balls indicate the Ca, C and O atoms respectively. The symbols ("+" and "−") inserted in **b** and **d** indicate the stacking sequences of carbonate layers introduced in Supplementary Fig. 8.

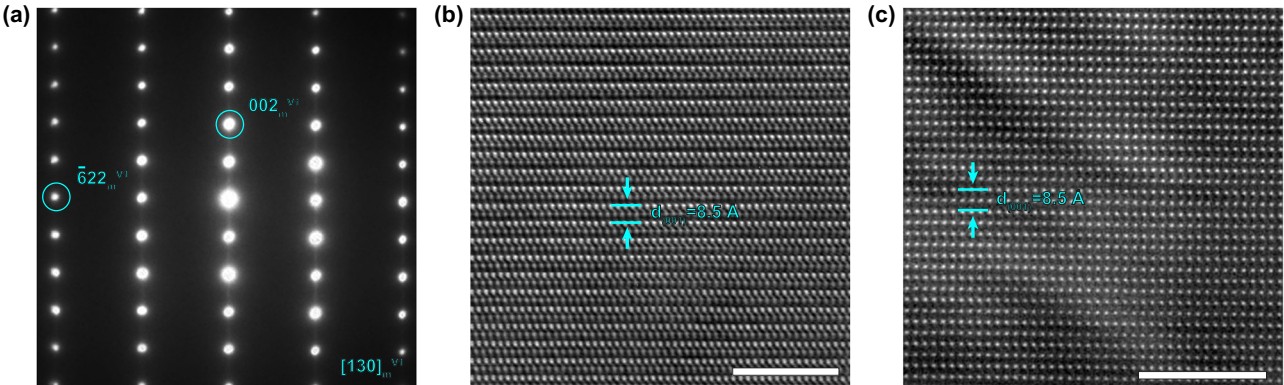

**Fig. 5 | Structural projection of vaterite along $[130]_m$ direction. a** SAED patterns. **b** Atomic resolution TEM image. **c** Atomic resolution HAADF image. Scale bar in **b** and **c** is 5 nm.

Figs. 15 and 16d. Additionally, we obtained structural segments with the same features as the structural projection of the monoclinic lattices along the $[010]_m$ and $[110]_m$ directions, supporting our conclusion from Fig. 4b. We also estimated the total energy of different stacking structures at the same temperature to understand the absence of the stacking sequence "0". The results indicate that the stacking sequence containing "0" significantly increases the potential energy of the system by 41 *meV per* $CaCO_3$ unit compared to those containing only "+" and/or "−" (See Supplementary Note VII and Supplementary Fig. 19). Therefore, any structure with the stacking sequence of "0" is energetically unfavorable. The EDPs simulations based on the theoretically grown structure shown in Fig. 4d agree well with the experimental observations along the stacking direction, namely, the $[103]_m$ zone axis (See Supplementary Fig. 20).

After tilting the grain by an additional 30° along the $(001)_m^*$ direction, the resulting $[130]_m$ zone-axis is obtained. As depicted in Fig. 5a, the SAED patterns indicate that the weak spots and diffused streaking feature between $(001)_m$ and $(002)_m$, which were observed in EDPs along the $[010]_m$ direction (Fig. 4a), are almost absent. Additionally, the atomic resolution TEM (Fig. 5b) and HAADF images (Fig. 5c) do not exhibit any evident stacking disorders. This can also be rationalized by the concept of orientation variants, where tilting variant I to the $[130]_m$ direction results in variants II and III being tilted to $[100]_m$ and $[1\bar{3}0]_m$, respectively. The simulated EDPs along these three

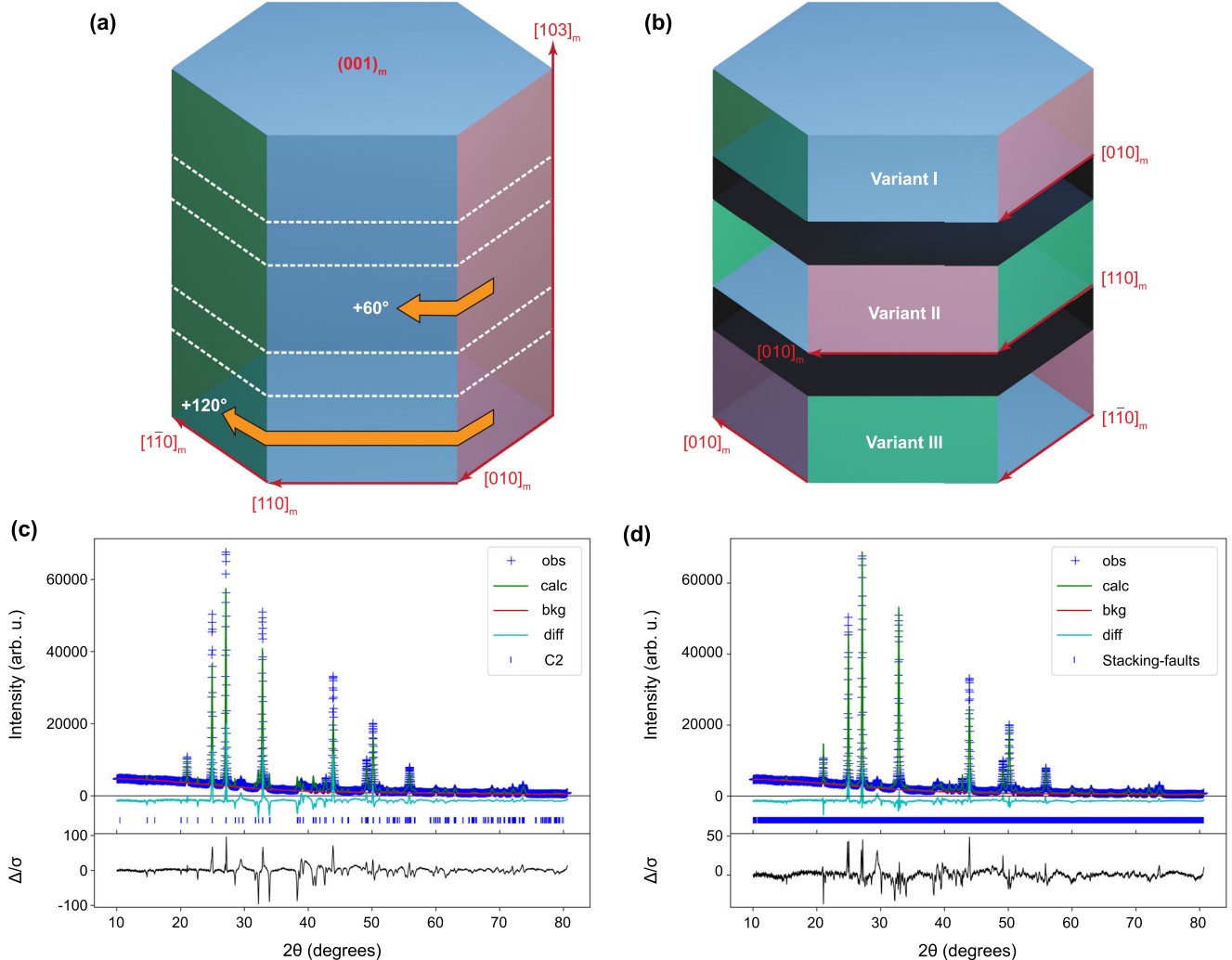

**Fig. 6 | Polytypic features within vaterite and XRD Rietveld refinement.** Schematic illustration showing the formation of the orientation variant. **a** Pristine structure with uniform orientations. **b** Polytypic structure with variants. Variants II and III can be obtained by continuous 60° rotation of the carbonate layers from variant I along the [103]$_m$ direction. The dark area in **b** indicates the disordered stacking between different variants. Rietveld refinement of the experimental

powder XRD pattern collected at room temperature against the **c** C2 model and **d** proposed polytypic model that contains stacking faults and orientation domains. Experimental data points are shown in cross-symbols, the calculated powder pattern is shown in green, the background is shown in red, and the difference curve between the experimental and calculated data is shown in blue color.

orientations are identical (See Supplementary Fig. 21). By overlapping the EDPs along these three orientations, the experimental observations shown in Fig. 5a can be reproduced.

Both experimental and theoritical results demonstrated that vaterite should be regarded as a polytypic stcuture. These variant segments, together with the stacking disorder, are interrelated along the [103]$_m$ direction, as shown schematically in Fig. 6a, b. As indicated in Supplementary Note VIII, although synchrotron X-ray and neutron diffraction are powerful toolsets, compared to conventional labaratory XRD, they do not show evident advantages in resolving the heavily faulted vateirte structure. Here, we performed the Rietveld refinement analysis of the powder XRD data against the C2 model and calculated supercell model introduced in Fig. 4d. The refinement results are shown in Fig. 6c, d. For C2 model, the agreement factor is around 21.88%. By introducing the stacking faults and orientation domains, the agreement factor is significantly improved to 12.07%. It should be noted that the supercell model used to calculate the theoretical XRD pattern is just one representative configuration obtained from a single molecular dynamics simulation. In other words, this supercell model does not encompass all possible stacking sequences of the bulk

materials. However, it still exhibits a reasonable match with the XRD data. Furthermore, from Fig. 6c, d, it is evident that our power XRD data accurately captures the presence of extra peaks, such as the ones with 2θ around 40°, which have been previously reported in other studies[42]. Clearly, the calculated XRD pattern based on our model explains these extra peaks well.

**Second-order phase transition in vaterite**

After clarifying the stacking principles of carbonate layers along the z-axis (See Supplementary Note VII and Supplementary Fig. 19), the question remains as to how the carbonate groups align in a single layer. Previous vaterite structure models[29,33], including the C2 structure, show that the carbonate groups tilt slightly away from their high-symmetry orientations. In addition, the slight distortion of the carbonate groups shows temperature sensitivity. Considering the important role of temperature for the nucleation, growth and phase transition of vaterite, it is essential to understand the temperature-dependent behavior of it. In addition, here, to unveil the real orientations of the carbonate groups in vaterite, we conducted a detailed analysis of their atomic positions, using the C2 phase as the initial structure. We

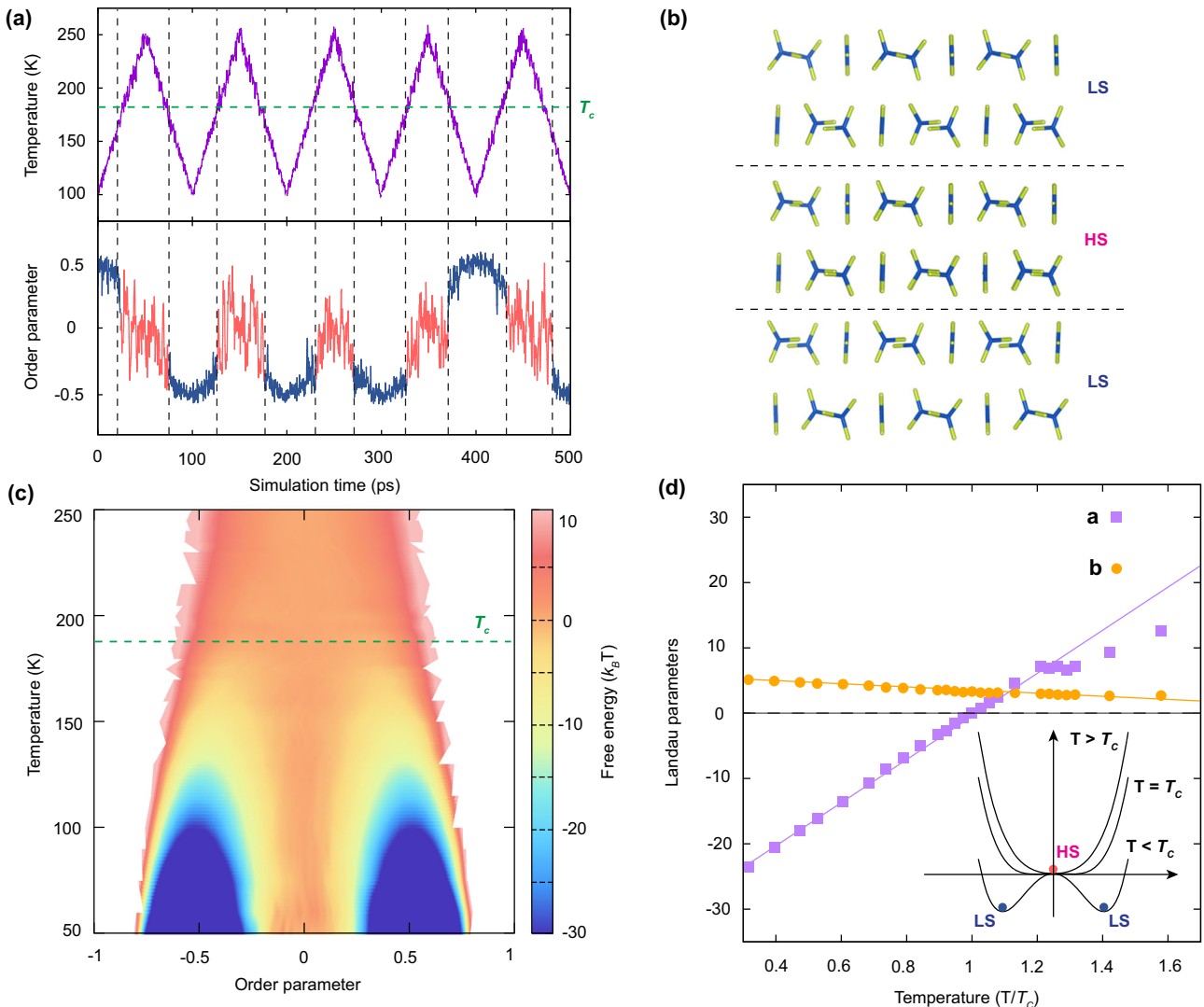

**Fig. 7 | Second-order phase transition in vaterite. a** Temperature and order parameter $Q$ as a function of simulation time. $T_c$ represents the critical temperature with the value of approximately 190 $K$. **b** Schematic illustration of the carbonate group orientations in high-symmetry (HS) and low-symmetry (LS) states. The carbonate group is shown in stick model, where carbon and oxygen atoms are colored with blue and yellow, respectively. **c** Two-dimensional free energy surface of vaterite as functions of order parameter $Q$ and temperature. **d** Landau parameters as a function of temperature. Inset: Ginzburg-Landau free energy surface below, at, and above the critical temperature $T_c$ of vaterite.

observed a small rotation of carbonate as the temperature increases, with the $C2$ phase transforming into the $C2/c$ phase at approximately 190 $K$, resulting in a higher symmetry. To quantify this phenomenon, we introduced an order parameter, $Q$, which measures the difference in the coordinates of the oxygen atoms in adjacent carbonate groups (See Methods and Supplementary Note XI). As shown in Fig. 7a, by periodically increasing/decreasing the temperature of the system, we find that $Q$ fluctuates around zero at high temperature, corresponding to a high-symmetry (HS) state (i.e., $C2/c$), and deviates away from zero at low temperatures, corresponding to a low-symmetry (LS) state (i.e., $C2$). Fig. 7b illustrates the orientations of carbonate groups in the HS and LS states. It is worth noting that, the other low-symmetry structure $Cc$ was not observed in our simulations, as it would transform into the $C2$ structure at a relatively low temperature due to its lower stability.

To understand the kinetic correlation between the two states, the well-tempered metadynamics[49], a smoothly converging and tunable free-energy method, was employed with the order parameter $Q$ serving as the collective variable. Fig. 7c plots the free energy surface as a function of $Q$ and temperature. At high temperatures, a parabola-like free energy surface is observed, while at low temperatures, an energy

barrier separates two local minima with the opposite order parameter value. This energy barrier decreases continuously with increasing temperature and eventually disappears at the critical temperature $T_c$ (190 ± 10 $K$), consistent with the findings presented in Fig. 7a. At the same time, the order parameter decreases continuously to zero, indicating a second-order phase transition in the Ginzburg-Landau model[50].

We find the free energy can be well described by $F(T, Q) = aQ^2 + bQ^4$, where $a$ and $b$ are phenomenological quantities known as Landau parameters. As predicted by the Ginzburg-Landau model, $a$ exhibits a linear temperature dependence near the critical point $T_c$, i.e., $a(T) = \alpha(T - T_c)$, while parameter $b$ is a constant. We estimated the Landau parameters by fitting the free energy surface in Fig. 7c and the results are shown in Fig. 7d. When the temperature is around $T_c$, both parameters can be well fitted. At relatively high temperatures, $a(T)$ starts to exhibit nonlinearity. Although $b(T)$ is not a perfect constant, the slope of its dependence on temperature is relatively small, and such characteristic has been reported in previous studies[51,52]. At $T_c$, the low energy area of the free energy surface becomes rather flat. We find that the LS structures with positive and negative $Q$ values are mirror-symmetrized to each other and share the same space group $C2$.

As demonstrated in Fig. 7a, the LS structures with different chirality can interconvert between each other passing through the HS structure. Such chirality feature in vaterite might be significant in some specific biosystems, in which one enantiomer might be favored[53].

## Discussion

In summary, a comprehensive investigation of biogenic vaterite from two different marine species, the lackluster pearls and the asteriscus of carp, was conducted by combining systematic TEM characterizations, crystallographic analyses, and advanced theoretical calculations. Eventually, the structural mystery of vaterite $CaCO_3$ is uncovered. The basic structure of vaterite has a monoclinic lattice with a space group of $C2$ or $C2/c$, with lattice parameters of $a = 12.3$ Å, $b = 7.13$ Å, $c = 9.4$ Å, and $\beta = 115.480°$. The difference between the $C2$ and $C2/c$ structures is characterized by the slightly different orientations of the carbonate groups within a single layer, which are sensitive to temperature, leading to different symmetries in vaterite. Importantly, the transformation between the low- and high-symmetry phases can be well described by a second-order phase transition with a critical point of approximately 190 $K$. Vaterite has a pseudohexagonal feature with a pseudo sixfold axis along the $[103]_m$ direction, and three variants that are formed due to a 60° rotation along this direction. In principle, vaterite is a polytypical structure formed by the nanoscale random intergrowth of the three orientation variants, which are interrelated along the $[103]_m$ direction. The stacking sequence "0" is absent, while "+" and "−" sequences are randomly distributed along the $[103]_m$ direction based on the monoclinic lattice. Additionally, the localized structural and chemical features of minerals are correlated with environmental nucleation and growth conditions, such as temperature, pressure and presence of impurities. However, the intrinsic structural features of vaterite are similar and the configuration rules do not change. The atomic scale insights of vaterite obtained from lackluster pearls and the asteriscus in the otoliths of carp are generally applicable to all the vaterite samples from different origination.

## Methods

### Materials and bulk characterizations

The lackluster pearl (vaterite structure) and normal pearl (aragonite structure) were collected from Zhuji, Southeast China. The asteriscus otolith pairs were extracted from the carps originated at Baiyang lake in China. Raman spectra were obtained in back-scattering mode at room temperature by utilizing the Jobin-Yvon HR800 micro-Raman system with a grating speed of 1800 *lines/mm*. This system was equipped with a liquid-nitrogen-cooled charge-coupled device (CCD) and a ×100 objective lens (NA = 0.9). The resolution of the Raman system was 0.35 $cm^{-1}$ per pixel. Raman spectra were measured using a 532 nm laser. Powder diffraction data were collected at room temperature on a STOE-STADI-P powder diffractometer equipped with an asymmetric curved Germanium monochromator (Cu $K_{\alpha1}$ radiation, $\lambda = 1.54056$ Å) and one-dimensional silicon strip detector (MYTHEN2 1K from DECTRIS). The line focused Cu X-ray tube was operated at 40 kV and 40 mA. Powder was packed in an 8 mm metallic mask and sandwiched between two polyimide layers of tape. Sample holder was span during collection. Intensity data from 10 to 80 degrees two theta were collected over a period of 12.5 h with an overall 0.005 degrees step. The instrument was calibrated against a NIST Silicon standard (640d) prior to the measurement.

### Electron transparent sample preparation and electron microscopy methods

The bulk vaterite samples were cut into different sections using a linear precision saw with a thickness of ~500 μm and then ground to ~100 μm using silicon carbide (SiC) papers. Thin foils with a diameter of <3 mm were stuck on the molybdenum grid, then grounded using variant grit SiC papers, and polished with diamond paste to ~10 μm and finally thinned by Ar ion milling in a Gatan precision ion polishing system (PIPS) using a low voltage (3–4 kV) to avoid possible ion beam damage. Furthermore, to improve the electronic conductivity, perforated samples were sputtered with carbon layer using a sputter machine (Quorum Q 150 R ES) for ~20 s.

A tilt series of SAED patterns were recorded using a FEI Tecnai T12 BioTWIN at 120 kV with a $LaB_6$ thermionic gun. The tilting angles along the X/Y axis range from −70°/−35° to +70°/+35°. Atomic scale resolution TEM and HAADF images were acquired using a JEM ARM200CF microscope which was operated at 200 $kV$. This microscope was equipped with a cold field-emission gun, a probe corrector, and the dual silicon drift detectors (SDDs). The point resolution in scanning TEM (STEM) mode is ~78 pm. The convergence semi-angle for STEM imaging was ~22 mrad. The inner and outer acceptance semi-angles for HAADF imaging were approximately 90 and 250 mrad, respectively. Within this collection angle range, the intensity of images is dominated by incoherent $Z$ contrast. The area of a single detector is 100 $mm^2$ and the solid angle of this energy dispersive spectrum (EDS) system is approximately 1.8 steradian ($sr$).

Simulations of the EDPs are performed using the CrystalDiffract software. Atomic resolution HAADF image simulations were performed using the Dr. Probe software[54]. The simulation parameters were chosen close to the experimental conditions assuming that all aberrations have been corrected within the probe forming aperture corresponding to a 21 mrad semi-convergence angle. The annular detector collection angles range from 80 to 200 mrad. An effective Gaussian source profile of 0.1 nm full width at half maximum (FWHM) was utilized. Simulated images were obtained as series over sample thicknesses up to 10 nm.

### Theoretical calculations

**Ab initio molecular dynamics calculation.** The ab initio molecular dynamics (AIMD) calculations were performed using the Vienna ab initio package (VASP)[55,56]. The general gradient approximation of Perdew–Burke–Ernzerhof (GGA-PBE) was adopted for the exchange-correlation functional[57,58]. The electron wave function was expanded using a plane wave with a cut-off energy of 400 eV. Reciprocal space integration was performed by setting the $k$ spacing to 0.5[59]. The AIMD was carried out with an isothermal-isobaric (NPT) ensemble at the temperature of 1000 K using the Langevin thermostat[60] and at the pressure of one bar using the Parrinello–Rahman barostat[61,62]. The time step in the simulation was 1 fs.

**Density functional theory (DFT) energy and force calculations.** Single point energy and forces were calculated using VASP. The smearing parameter and exchange-correlation density functional were the same as those in the AIMD simulation, while the energy cut-off for this calculation was increased to 1000 eV.

**Training deep neural network model.** All training of calcium carbonate DNN models was performed with the package DeePMD-kit[45]. The settings of the training parameters are as follows: the cut-off radius smoothly decays from 7.5 Å to 8 Å. The sizes of three hidden layers for the embedding network and three hidden layers for the fitting network were set to (20, 40, 80) and (240, 240, 240) respectively. The learning rate decays from $1.0 \times 10^{-3}$ to $5.3 \times 10^{-8}$. The pre-factors of the energy, force and virial term in the loss function change from 0.04 to 0.2, 1000 to 1 and 0.04 to 0.2, respectively. The total number of training steps was set to $1 \times 10^6$.

To establish a training set capable of covering the whole solidification process, first, we collected configurations from the AIMD trajectories, in which we fixed half of the calcium atoms in the supercell while carbonate ions were allowed to rotate freely at high

temperatures to obtain various liquid–solid interface structures. Approximately 13,000 configurations were obtained together with the corresponding energy, force, and virial data as the training set to establish the first-generation DNN model (DNN1). We next performed MD simulations on Lammps[63] using DNN1 to enable adequate sampling of the configuration space at larger timescales. To make our DNN model accurately predict the energies and forces of the structures under various thermodynamic conditions, the MD simulations covered the temperature range 1–2000 K. Additionally, initial solid structures with different stacking sequences along the z-axis were also considered. Notably, the orientations of carbonates varied casually at high temperature before melting, which indicated that not only the structures with ordered carbonates distributed over three orientations in one layer, but also the structures with disordered carbonates were included in our training set.

Based on the configurations obtained in the previous steps, we performed DFT single point energy calculations and constructed an updated training set to establish DNN2. We repeated the above steps to perform MD simulations under the corresponding thermodynamic conditions and calculated the single point energies for the configurations extracted from the trajectories. The energies and forces calculated by DFT were compared with the predicted values of DNN (See Supplementary Fig. 17), and the configurations with large error were added to the training set for iterative optimization. The errors of the final DNN model in terms of the atomic energies and forces on the test sets were $1.81 \times 10^{-3}$ eV/atom and 0.11 eV/Å respectively.

**Molecular dynamics simulations.** The molecular dynamics simulations were carried out using Lammps, integrated with DeePMD-kit to utilize the DNN potential. The simulation system with 6480 atoms (1296 $CaCO_3$ units), consisting of 36 layers of calcium atoms and 36 layers of carbonates, was in an orthorhombic box with periodic boundary conditions in all three directions with the dimensions of $2.2017 \times 2.5406 \times 15.3321$ nm$^3$.

To perform the following crystallization simulations, we first constructed a liquid–solid coexisting configuration in which the solid phase was composed of only one single layer of vaterite. This was done through a simulation at a high temperature of 1600 K using the NPT ensemble starting from a perfect structure of vaterite, in which one layer of calcium atoms was fixed. Thus, the ordered calcium carbonates transformed into disordered states except for the fixed layer of calcium atoms. Since calcium atoms exhibit a hexagonal crystal structure in vaterite, fixing a layer of calcium atoms would allow the system to grow more easily in the specific direction. Then, we used the obtained configuration to perform the crystallization simulations. The liquid–solid phase transition proceeded along the [001] direction of the hexagonal lattice during 1000 K relaxation of the NPT mode. Our structure arises from the natural transformation kinetics and therefore indicates the intrinsic properties of the structure of vaterite alone the [001] direction. After obtaining a complete solid phase, we further quench the system to 10 K in 1 ns to reduce the thermal fluctuations. Snapshots of the trajectory are presented in Supplementary Fig. 18. Several simulations under the same thermodynamic conditions, but the velocities with different random seeds, were performed to cross-validate the simulation results.

In the above simulations, temperature and pressure were controlled by the stochastic velocity rescaling thermostat[64] and the Parrinello–Rahman barostat, respectively. The timestep is set to 1 fs. The initial velocity of each atom was generated with the Boltzmann distribution. The linear momentum was rescaled by subtracting the momentum of the center of the mass every 1000 timesteps. Plumed[65] packaged in Lammps was used for performing well-tempered metadynamics[66] and analyzing the trajectories. Crystal structure visualization was achieved using Ovito[67] and Vesta[68].

**Well-tempered metadynamics and free energy surface.** To estimate the difference in free energy between the high symmetry (HS) phase and the low symmetry (LS) phase at different temperatures, we further performed well-tempered metadynamics (WTMetaD) with the order parameter Q introduced in Supplementary Note IX.

The bias potential of WTMetaD is a history-dependent function of the order parameter composed by the deposits of Gaussians, and the height of the additive Gaussian dwindles as the sampling proceeds. The comprehensive descriptions were listed in previous work[66]. In this work, the width and the initial height of Gaussian are set to 0.075 Å and 3 kJ/mol, respectively. The bias factor, which characterizes the rate of change of the deposited Gaussian height, is set to 30. The time interval of the deposits of Gaussians is equal to 500 fs. We calculated the free energy surfaces at 24 different temperatures with a temperature interval of 10 K from 50 K to 250 K. The other MD settings for the simulations were the same as those used in the growth process as described above. The simulation time was set as $2 \times 10^6$ steps (2 ns) to reach convergence. After the sampling was finished, we reweighted the obtained data using an on-the-fly strategy[69] to evaluate the free energy surfaces.

## Data availability
All data needed to evaluate the conclusions of this work are presented in the paper and the Supplementary Materials. Additional data related to this paper may be requested from the authors.

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

## Acknowledgements
X.S. acknowledges the National Natural Science Foundation of China (No. 51901065), the Nature Science Foundation of Hebei Province (No. E2020201023), and the Advanced Talents Incubation Program of Hebei University (No. 521000981164) for financial support. H.N. was supported by the National Science Fund for Excellent Young Scientist Fund Program (Overseas) of China, the National Natural Science Foundation of China (No.92370118), the Science and Technology Activities fund for Overseas Researchers of Shaanxi Province, and the Research Fund of the State Key Laboratory of Solidification Proceeding (NPU) of China (No. 2020-QZ-03). This work made use of the EPIC facility of Northwestern University's NU*ANCE* Center, which has received support from the Soft and Hybrid Nanotechnology Experimental (SHyNE) Resource (NSF ECCS-2025633), the International Institute for Nanotechnology (IIN), and Northwestern's MRSEC program (NSF DMR-2308691). This work also made use of the Integrated Molecular Structure Education and Research Center (IMSERC) at Northwestern University, which has received support from the State of Illinois, Northwestern University, ShyNE Resource (NSF ECCS-2025633), NSFCHE-1048773, and the IIN. The authors are grateful to Prof. Qingling Feng at Tsinghua University for kindly providing the bulk pearl samples and Dr. Chi Zhang in Ocean University of China for many helpful discussions on vaterite structures.

## Author contributions
Conceptualization: X.H., H.N.; Methodology and Analysis: X.S., J.H., M.C., H.N., C.M., X.H.; All authors contributed to the discussion and writing.

## Competing interests
The authors declare no competing interests.
