## [Peer Review File · Nature Communications]

Reviewers' Comments:

Reviewer #1:

Remarks to the Author:

Dear Authors,

This work is very interesting. It is intended to contribute to complete the structural and phase transformation descriptions of vaterite, this latter having been matter of large debates over many decades.

I think that I am missing few experimental evidences in order to be able to accept your views concerning disorders in vaterite, which prevent me from giving out a clean opinion.

All vaterite samples I could see up to now, biogenic or pure syntheses, exhibit diffraction signatures with extra peaks out of any "regular" modeling. Regular means here using classical 3D space groups, including faulted samples. This has been the purpose since the beginning of this long debate.

In your TEM measurements you are indicating "diffuse" scattering between main peaks, the ones that you faulted model are aiming at reproduce. However, in the shown TEM diffraction patterns (Fig 3) such signatures are not so much visible, at least not enough to see if these "diffuse signals" are indeed diffuse or peaky, eventually coming from aperiodic peaks, commensurate or not. I mean, is your approach able to reproduce such signatures correctly or not, using usual space group + disorders and faults ? Unfortunately, the promised XRD diagram is not provided (or I have not been able to find it).

On the other hand, the work by Steciuk et al. (ref 27 in the main article, or 11 in the supplements) clearly is able to fit all these extra peaks, observed also in the precession images, using aperiodic space group formalisms. Some of your HRTEM images clearly show atomic modulations that very probably could be understood using such 3D aperiodicities, in space groups larger than 3-dimensional.

Then the main issue I would like to be fixed before considering acceptance of this work would be to see on XRD and/or TEM diffraction patterns if the proposed models are able to reproduce the extra peaks observed by so many authors and which created the vaterite conundrum ! If this is the case, then we could very interestingly compare your approach with superspace descriptions

I think you haven't read enough carefully Steciuk's work: it is not concentrated on the C2/c space group, but on its aperiodic variants in superspace able to reproduce the extra peaks. This is why I insist on this. Please provide 1D fitted patterns in which we see the intensities reproduction using your faulted models.

Last comment: using simulations, you justify the absence of so-called "0" variants on the basis of a 41 meV/f.u. difference. I was wondering if such a "small" magnitude is reachable using the simulations used, and if yes, what would be the associated uncertainties ? Aragonite is at +1 kJ/mol from calcite in normal conditions, and vaterite at much more, then presumably 41 meV/f.u. can be considered small enough not to consider full absence of "0" variants ?

This work is potentially of large impact, but I really need to be able to see how the proposed faults are ameliorating diffraction fits.

Sincerely,

D. Chateigner

Reviewer #2:

Remarks to the Author:

It is a solid piece of work on a very important type of material, using advanced imaging methods that are supported by MD simulations. Machine learning was applied as well. The Supplementary Notes are really good and they make a fascinating read, showing at the same time, the depth that authors have gone to trying to understand the challenges of defining a crystal structure of vaterite.

I do, however, have issues with the way the results are presented and more importantly not extending them to the results obtained by other methods. Synchrotron x-ray and neutron diffraction are the methods specifically developed for solving complex crystal and magnetic (in the case of neutrons) structures. Both are volume-averaged techniques that can help to understand to what extent the findings about stacking faults obtained with a local probe such as HRTEM and HAADF are applicable to the entire sample. In all the applications that the authors mentioned in the introduction, a substantial volume of vaterite will be used. How do your results map on the samples of several cubic cm? Neutrons are sensitive to light elements like O and C, providing a much better contrast than lab XRD. The authors had a discussion in the SI about difficulties to understand lab XRD results and briefly dismissed synchrotron x-rays results as well. It is puzzling because synchrotron x-rays provide superior resolution, small background. Using a 2D position-sensitive detector can show you presence of the crystallites in the powdered samples. X-ray diffraction measurements on a single nanoparticle are now possible, not to mention anomalous x-ray scattering can be done as well. Yet, all the scattering results are dismissed based on lab XRD work.

I would like to see an extended discussion on why neutron and scattering methods could not be applied to vaterite. There are many good examples of how microscopy and scattering results complement each other (<https://doi.org/10.1038/ncomms6964>, [10.1039/D0NR08615K](https://doi.org/10.1039/D0NR08615K), just to name a few). Did the authors do the Rietveld analysis of their XRD data presented in Ext. Fig. 1b? Did you try to simulate x-ray and neutron diffraction patterns can compare them with the literature? This is a necessary piece, which is missing.

More specific comments:

p.1, Abstract, first sentence: "... plays a critical role in various fields". What fields? Fields of applications or research?

p.3: Why these two particular samples were chosen for this work? What is so special about them? Please, elaborate on this topic the Intro section.

p.9: Why do you claim that the obtained structural information is intrinsic to all vaterite samples? Is this conclusion based on an analysis of just two samples? Please, elaborate more. The same sentence is repapered in the Conclusions.

p.11: Why the temperature-dependent studies are important? What is your motivation? In the Intro section, most of the applications you mentioned operate at room temperature. Please, explain why temperature-induced phase transitions are important for vaterite.

Reviewer #3:

Remarks to the Author:

The paper presents evidence for vaterite being a polytype rather than a defined structure. This idea has been out for nearly a century, and proven and consolidated through both experiments and theory in the past decade:

- Meyer, Z. *Kristallogr.* 1969, 128, 183 (XRD): hypothesis for stacking faults
- Qiao & Feng, J. *Cryst. Growth*, 2007, 304, 253 (HRTEM & SAED): twin stacking faults observed
- Wehrmeister et al. *J. Raman Spectrosc.* 2010, 41, 193 (Raman spectroscopy): hypothesis for polytypism, together with evidence for more than 2 independent CO₃²⁻ in the structure
- Demichelis et al. *CrystEngComm* 2012, 14, 44; *Cryst. Growth. Des.* 2013, 13, 2247 (DFT): re-analysing the whole set of experimental and theoretical structures, excluding the some of the proposed structures, finding others more compatible to experimental evidence and theoretical investigations, and providing evidence that different polytypes do not differ significantly in energy;
- Kabalah-Amitai et al. *Science* 2013, 340, 454 (HRTEM): at least 2 distinct structures for vaterite identified in *Herdmania momus* spicules, given as hexagonal and 1 undetermined; the main differences between the structures are related to CO₃²⁻ (as discussed in Wehrmeister et al 2010 and in Demichelis et al 2012, 2013 and in all the literature after then). Here it is true that the hexagonal space group selected does not explain any of the weak diffractions, however, this has

been addressed in the literature from 2012 on;

-Balan et al. *Chemical Geology*, 2014, 374–375, 84 (QM calculation): mixing energies of sulfate into vaterite are consistent with one of the hexagonal models discussed above (one of the two polytypes, P3221);

- Burgess & Bryce, *Solid State Nucl. Mag. Res.* 2015, 65: 75-83 (43Ca NMR, XRD): two structures (polytypes P3221 and C2) compatible with recorded 43Ca NMR spectra and both compatible with XRD;

- De La Pierre et al. 2014, *J. Phys Chem C* 2014, 118, 27493 (experimental and DFT Raman spectroscopy):two structures (polytypes P3221 and C2)have computed Raman spectrum that is compatible to experimental Raman spectra (referred to as multiple structure there)

- Christy, *Cryst. Growth Des.* 2017, 17, 6, 3567–3578: reexamination of the whole literature, including the above references, and conclusions that it's a case of multiple structures/polytypism, with further exclusion of some theoretical structures based on crystallography.

After Christy's analysis, there is no longer a debate on the fact that vaterite exhibits polytypism/multiple structure. While it is true that some sort of doubt still exist about the actual space groups that can describe vaterite, and more images/evidence about its polytypism are welcome, many of the "unknown" features claimed in this paper have already been addressed.

While there is value in having further evidence that supports vaterite polytypism, this paper does not match the novelty requirement of *Nature Communication*, and it does not contain any important advance of significance to specialists within the fields related to vaterite (mostly geochemistry, biology, chemistry). The authors should consider submitting this paper for consideration to a more specialised journal (e.g. *J. Cryst. Growth*) after reworking introduction and discussion through adding more details about the actual findings in the aforementioned literature.

Reviewer #4:

Remarks to the Author:

A valuable contribution to understanding the complex real structure of vaterite, which is increasingly being recognised as an important biomineral.

I have made some comments and corrections in notes on a pdf of the manuscript, which is attached.

Unlocking the Structural Mystery of Vaterite CaCO₃

Xingyuan San^{1†}, Junwei Hu^{2†}, Mingyi Chen², Haiyang Niu^{2*}, Paul J. M. Smeets³, Jie Deng⁴,
Kunmo Koo³, Roberto dos Reis³, Vinayak P. Dravid^{3*}, Xiaobing Hu^{3*}

¹Hebei Key Lab of Optic-electronic Information and Materials, The College of Physics Science and Technology, Hebei University, Baoding 071002, China

²State Key Laboratory of Solidification Processing, International Center for Materials Discovery, School of Materials Science and Engineering, Northwestern Polytechnical University, Xi'an 710072, China

³Department of Materials Science and Engineering, The NUANCE Center, Northwestern University, Evanston, IL 60208, USA

⁴Department of Geosciences, Princeton University, Princeton, NJ 08544, USA

[†]These authors contributed equally to this work.

*Corresponding authors. Email: xbhu@northwestern.edu (X.H.); haiyang.niu@nwpu.edu.cn (H.N.); v-dravid@northwestern.edu (V.P.D.)

Abstract

Calcium carbonate (CaCO₃), the most abundant biogenic mineral on earth, plays a crucial role in various fields. Of the four polymorphs, calcite, aragonite, vaterite, and amorphous CaCO₃, vaterite is the most enigmatic one due to an ongoing debate regarding its structure that has persisted for nearly a century. In this work, based on systematic transmission electron microscopy characterizations, elaborate crystallographic analysis and machine learning aided molecular dynamics simulations with *ab initio* accuracy, we reveal that vaterite can be regarded as a polytypic structure. The basic phase  monoclinic lattice possessing pseudohexagonal symmetry. Direct imaging and atomic-scale simulations provide evidence that a single grain of vaterite can have three orientation variants . Additionally, we find that vaterite undergoes a second-order phase transition. These atomic scale insights provide a comprehensive understanding of the structure of vaterite and offer new perspectives on the biomineralization process of calcium carbonate.

Introduction

Calcium carbonate (CaCO_3) is one of the most abundant natural materials found in biomineral systems, which plays a pivotal role in the chemistry of the hydrosphere, lithosphere, biosphere, climate regulation, and scale formation (1-4). Among the three anhydrous crystalline structures, vaterite (5), aragonite (6) and calcite (7), vaterite has the most confined stability field. It is frequently observed as a transient intermediate phase during the transition process of amorphous calcium carbonate to other more stable polymorphs (8, 9) or formed *via* liquid–liquid phase separation in supersaturated CaCO_3 solution (10-12), thereby kinetically directing the mineralization pathway. Although vaterite is less stable and rarer compared to other forms of CaCO_3 due to its relatively low stability, it can still be commonly found in many biogenic and abiotic systems, such as freshwater lackluster pearls (13), cement (14) and natural minerals (15). In addition, vaterite's unique structure, high solubility in water, and porosity properties relative to other crystalline polymorphs make it an ideal functional material for environmental chemistry (16, 17), bone tissue (18, 19) and biomedical engineering (5, 20). Therefore, an accurate description of the structure of vaterite at the atomic scale is very demanding for gaining the rationale behind the growth dynamics of calcium carbonate (10, 21-23) and fulfilling its potential applications as a functional material (24-26).

Despite decades of experimental and theoretical efforts (27-31), the precise structure of vaterite remains a subject of debate (Supplementary Note I, Table S1). Several conflicting structures have been proposed, such as the hexagonal Meyer lattice (32), orthorhombic lattice (33, 34), monoclinic lattice and triclinic lattice (29, 30). The discrepancies arise not only from the ordering of carbonate groups, particularly the slight rotation within a single layer, but also from the long-range stacking sequence of carbonate layers along the *z*-axis concerning a hexagonal lattice. Recently, by means of aberration-corrected high-resolution transmission electron microscopy (HRTEM), Pokroy *et al* revealed that vaterite crystals contain two interspersed crystal structures, namely a predominant hexagonal Kamhi lattice together with an unknown structure. Although a combined Kamhi and Meyer model can be used to explain the strong diffraction spots, neither one can explain any of the weak diffractions (31). Thus, the real structural features of vaterite are still a mystery to humankind. Until now, it has been generally acknowledged that vaterite cannot be described using a single lattice and there are likely many planar faults including

micro twinning and other stacking orders and/or disorders (27, 33, 35). However, to our knowledge, direct imaging evidence of various stacking disorders at the atomic scale in vaterite is still missing.

Uncovering the structural features of vaterite through experimental means is an extremely challenging task. Conventional bulk measurements such as X-ray diffraction (XRD), electron diffraction and Raman spectra are not sufficient, and atomic resolution imaging is a prerequisite. HRTEM imaging is one choice, however, this technique is actually very limited in resolving the complex structure here since it can only provide a phase contrast image that usually contains many artifacts caused by thickness and focus variations. In contrast, atomic resolution high angle annular dark field (HAADF) imaging is an ideal technique since it provides the atomic number (Z) contrast with the intensity in proportion to $Z^{1.7-1.9}$ (36). Here, we uncover the structural mystery of vaterite by combining atomic resolution experiments based on samples from two different sources, the lackluster pearls and the asteriscus of carp, and molecular dynamics simulations adopting a deep neural network (DNN) interatomic potential with *ab initio* accuracy. The results of the two paradigms are complementary and can be cross validated. Our atomic scale insights into the vaterite structure resolve the contentions that existed among previous research work and provide an intrinsic perspective for studying the biomineralization mechanism of CaCO_3 , which may consequently facilitate the synthesis of calcium carbonate based materials with target structure and performance.

Fig. 1. Microstructural features of vaterite. (a) Splitting of the most intense Raman band ν_1 . The best result of the Gaussian fitting procedure was achieved for the decomposition of three peaks at 1075 cm^{-1} , 1080 cm^{-1} and 1091 cm^{-1} . (b) HAADF image showing the laminate structure. The inset shows the morphological features of the pearls with vaterite structure. (c) HRTEM image and inset FFT patterns showing the representative faulted features of vaterite.

Results and discussion

General structural features of vaterite

We first measured the Raman spectra of the lackluster pearls. As shown in Fig. 1a and Extended Data Fig. 1a, the splitting Raman band ν_1 , which represents the symmetric stretching mode of the carbonate ions, has the highest intensity. The spectral features shown here are a good match to those obtained from geological, synthetic and other biogenic samples comprising vaterite minerals (37-39), indicating that our lackluster sample can be categorized as a common biomineral containing a vaterite structure. The most intense peak is at 1091 cm^{-1} , with consecutively lower intensities for the peaks at 1075 cm^{-1} and 1080 cm^{-1} . The triple splitting suggests the presence of at least three crystallographically independent carbonate groups and similar carbonate group layers within vaterite (40, 41). In addition, as shown in Extended Data Fig. 1b, our XRD data agree well with other diffraction data of vaterite in the literature (37-39), indicating again that our lackluster pearls are representative biominerals containing a vaterite structure.

Fig. 2. A series of SAED patterns of vaterite obtained from a single grain. EDPs along the (a) $[103]_m$, (b) $[001]_m^{V1}$, (c) $[101]_m^{V1}$, (d) $[001]_m^{V2}$, (e) $[101]_m^{V2}$ and (f) $[001]_m^{V3}$ directions. The

subscript m represents the monoclinic lattice. The superscripts $V1$, $V2$ and $V3$ indicate three orientational variants. Circles in (a) indicate two sets of reflections with significant differences in intensities with Set I/II patterns having higher/lower intensities.

Figure 1b presents a high-angle annular dark field (HAADF) image of the lackluster pearl, which shows a laminar structure at the micrometer scale. The associated elemental maps, shown in Extended Data Fig. 1c-e, confirm the uniform distribution of C, Ca and O in the grain interior. Additionally, the HRTEM image in Fig. 1c reveals the general stacking feature of vaterite at the atomic scale. The inset digital fast Fourier transformed (FFT) patterns suggest the presence of many disordered stacking layers along the $(001)_H$ plane, labeled based on the hexagonal Meyer lattice (32). This finding is consistent with previous results on other vaterite samples from different originations (30, 31, 35). However, due to the significant offset resulting from the phase contrast, the HRTEM image in Fig. 1c cannot be used to extract further structural details.

To uncover the intrinsic structural features of vaterite, we then performed large angle tilting experiments to obtain serial selected area electron diffraction (SAED) patterns from a single grain. The relative positions and associated experimental tilting angles of each diffraction pattern in reciprocal space are given schematically in Supplementary Fig. S1, and diffraction patterns are displayed in Fig. 2. The diffraction patterns in Fig. 2a reveal a hexagonal symmetry feature, which is further confirmed by convergent beam electron diffraction and atomic resolution imaging analysis (Extended Data Fig. 2). Such results agree well with the suggested the hexagonal structure (42, 43). However, the diffraction patterns in Fig. 2b-f can only be indexed by a monoclinic structure with lattice parameters of $a=12.2 \text{ \AA}$, $b=7.2 \text{ \AA}$, $c=9.3 \text{ \AA}$ and $\beta=115.2^\circ$. These lattice parameters could match any of three monoclinic lattices including the $C2$, Cc , and $C2/c$ structures proposed in previous work (29). Unfortunately, diffraction data (Supplementary Fig. S4, S5 and Note III) cannot distinguish between the different rotations of carbonates of these three structures. The structural projection of the $C2$ structure along the $[010]$ direction is illustrated in Extended Data Fig. 3a, which can also be described using a hexagonal lattice as highlighted by the dashed line. In fact, the aforementioned monoclinic lattice is closely related to the disordered Meyer structure with a hexagonal lattice ($P6_3/mmc$; $a=b=7.169 \text{ \AA}$, $c=16.98 \text{ \AA}$). Their intrinsic relationships are discussed in Supplementary Note IV. As shown in Extended Data Fig. 3b, calcium ions (Ca^{2+}) approximately form a hexagonal array, and carbonate ions (CO_3^{2-}) occupy half

of the Ca_6 trigonal prismatic interstices in one layer and the other half in the neighboring layers. In each interstice, the carbonate has three possible orientations, and the carbonate ions within a plane are regularly distributed resulting in an ordered arrangement that is more energetically favored (44). Considering the rotational invariant, the stacking sequence composed of two adjacent layers is always identical (Supplementary Fig. S10). Once a third layer is introduced, three possible stacking sequences can be obtained, *i.e.*, the carbonate groups in the third layer with three possible orientations correspond to rotations of $+120^\circ$, -120° , or 0° around the z -axis compared to the counterparts in the first layer (Supplementary Fig. S11). The three stacking sequences can be marked as “+”, “-”, and “0”, respectively, following the notation originally proposed by Christy (28) (Supplementary Note II and Extended Data Fig. 3c). The $C2$ phase shown in Extended Data Fig. 3a and its two analogous structures, namely the Cc and $C2/c$ structures share the same stacking sequence (+--+--+), only the rotation of carbonates is slightly different.

In Fig. 2a, the electron diffraction patterns (EDPs) show two sets of reflections with different intensities, as indicated by circles. Only the spots with brighter contrast (Set-I patterns) match the monoclinic structure along the $[103]_m$ direction. In contrast, all of the reflections in Fig. 2b-f can be solely indexed along either the $[001]_m$ or $[101]_m$ direction based on the monoclinic lattice, despite the obvious intensity difference. Importantly, the relative distributions of the reflections in Fig. 2b, 2d and 2f are the same, except for a 60° rotation along the viewing direction. This phenomenon is also observed in Fig. 2c and 2e, indicating that there are likely three orientation variants of the monoclinic lattice within the vaterite structure (Supplementary Fig. S6 and Note V). These three variants can be formed by a 60° rotation along the $[103]_m$ zone-axis, considering the pseudo-hexagonal features of the monoclinic lattice. The stacking of variant segments along the $[103]_m$ direction yields the stacking disorder observed in Fig. 1c and the weaker spots in Fig. 2a-2f (Supplementary Note VI). Therefore, the domain concept proposed here can well rationalize the broadly existed weak spots that could not be explained in previous work (31). Direct evidence of orientation variants in real space will be discussed below.

Localized structural intergrowth within vaterite

Fig. 3. Similarity and nuance difference for the stacking of carbonates along some directions.

Structural projection of the ordered $C2$ vaterite along the (a) $[110]_m$, (b) $[1\bar{1}0]_m$, and (c) $[010]_m$ directions. The black frames indicate the projection of a unit cell along the corresponding directions. The blue vertical lines indicate the stacking feature of the Ca-C-O chains along the $[001]_m$ direction. Atomic resolution HAADF image simulations along the (d) $[110]_m$, (e) $[1\bar{1}0]_m$, and (f) $[010]_m$ directions. The red circles in (d-f) indicate the diffuse streaks along the vertical direction.

To visualize the potential stacking order and/or disorder within vaterite, it is necessary to tilt the grain to the $[010]_m$ zone-axis, as this direction allows for edged-on projection of the planar defects. To facilitate experimental data analysis, we would like to uncover the slight difference along some $[010]_m$ related directions first. As demonstrated in Supplementary Fig. S6 schematic, the monoclinic lattice has a pseudo-hexagonal feature with a pseudo-6-fold axis along the $[103]_m$ direction. The $[010]_m$ direction is equivalent to the $[110]_m$ and $[1\bar{1}0]_m$ directions. Simulated EDPs along the above three directions are shown in Supplementary Fig. S9a-c, and structural projections of the above three directions are shown in Fig. 3a-c. It is observed that the stacking features of the Ca layers are nearly identical along the above three directions. In between two neighboring Ca layers along the vertical direction, there are layers composed of carbonates. However, along these projected directions, the stacking features of the carbonate differ (indicated by vertical lines).

Based on the atomic resolution HAADF image simulations shown in Fig. 3d-f, it is found that the C-O bonds orientated approximately along the vertical directions can generate the diffuse streak features indicated by red circles. Therefore, the stacking features of the diffuse streaks can be used to differentiate the stacking features of the carbonate layers.

Fig. 4. Stacking disorder within vaterite. (a) EDPs obtained from vaterite along the $[010]_m$ zone-axis. (b) Atomic resolution HAADF image along the $[010]_m$ zone axis showing the polytypic features within vaterite. The red vertical lines indicate the stacking feature of the Ca-C-O chains along the $[001]_m$ direction. Inset image simulations correspond to the vaterite structure with the $C2$ space group along the $[110]_m$ and $[010]_m$ directions. (c) Strain map (ϵ_{xy}) of the image shown in (b) obtained by geometric phase analysis (GPA). Inset FFT patterns corresponding to areas A. (d) Theoretically grown vaterite with polytypic structure. The symbols (“+” and “-”) inserted in (b) and (d) indicate the stacking sequences of carbonate layers introduced in Extended Data Fig. 3.

In Fig. 4a, the experimental EDPs of vaterite are shown along the $[010]_m$ zone-axis, indicating a significant amount of stacking disorder on the $(001)_m$ plane due to the strong streaking features observed on this plane. Figure 4b displays an atomic resolution HAADF image along this direction, while the corresponding strain map derived from the geometrical phase analysis (GPA) is shown

in Fig. 4c. The interface separating structures of different lattices are apparent, with the right and left regions labeled as A and B, respectively. The inset FFT patterns suggest that A can be indexed solely based on the monoclinic lattice along the $[010]_m$ direction. Within the vaterite grain interior, there are three variants, with variants II and III corresponding to the $[110]_m$ and $[1\bar{1}0]_m$ directions, respectively, if variant I is tilted to the $[010]_m$ direction. While structural projections along these three directions are very similar, as shown in Fig. 3, slightly different stacking sequences of carbonate layers, located between the two calcium layers, can be used to differentiate the variations in the local structural segment. Figure 4b shows that some regions correspond to the monoclinic structure projected along the $[010]_m$ direction, while others correspond to the structural projection of the monoclinic lattice along the $[110]_m$ direction. The atomic resolution HAADF image with a larger field of view, shown in Supplementary Fig. S12, clearly demonstrates that the vaterite structure should be regarded as a polytypic structure with three orientation variants formed based on the monoclinic lattice. These variant segments, together with the stacking disorder, are interrelated along the $[103]_m$ direction, as shown schematically in Extended Data Fig. 4.

It is worth noting that the atomic scale structural features obtained from lackluster pearls are reproducible in other vaterite samples of different originations. Specifically, we have compared characterization results with another vaterite biomineral, an asteriscus present in the otolith of carp, for further analysis. As demonstrated in Supplementary Data Fig. 5, the Raman spectrum of the asteriscus shows typical features of vaterite, and the HRTEM images and associated FFT patterns demonstrate the heavily faulted feature of vaterite grains, similar to what we found in the lackluster sample. Additionally, the atomic resolution HAADF image in Extended Fig. 5d shows the localized stacking features of calcium and carbonate layers, which can also be labelled based on the signs introduced in Extended Data Fig. 3. This direct evidence confirms that the primary structural information obtained from the lackluster pearl is the intrinsic information of vaterite and is generally applicable to other minerals containing a vaterite structure.

To validate our experimental results, we conducted molecular dynamics (MD) simulations to study the crystallization process of vaterite directly from liquid. We trained a DNN potential under the framework of the DeePMD approach (45), which can simulate a large system with *ab initio* accuracy for a long time (46-48), which is essential for crystallization simulation. To overcome the nucleation barrier, solid-liquid coexistence simulations (Extended Data Fig. 6) were performed,

starting with a single fixed Ca solid layer at 1000 K. The crystallization proceeded with the reorientation of the carbonate groups, and once completed, the system was quenched to 10 K to eliminate thermal fluctuations. One of the final configurations is shown in Fig. 4d. Our simulation results reveal that the stacking sequence “0” is absent, while “+” and “-” are randomly distributed along the z-axis, equivalent to the $[103]_m$ direction in the monoclinic lattice (normal direction of the $(001)_m$ plane). These results are in excellent agreement with our experimental findings shown in Fig. 4b, Supplementary Fig. S12 and Extended Data Fig. 5. Additionally, we obtained structural segments with the same features as the structural projection of the monoclinic lattices along the $[010]_m$ and $[110]_m$ directions, supporting our conclusion from Fig. 4b. We also estimated the total energy of different stacking structures at the same temperature to understand the absence of the stacking sequence “0”. The results indicate that the stacking sequence containing “0” significantly increases the potential energy of the system by 41 *meV per* CaCO_3 unit compared to those containing only “+” and/or “-” (Supplementary Note VII and Fig. S14). Therefore, any structure with the stacking sequence of “0” is energetically unfavorable. The EDPs simulations based on the theoretically grown structure shown in Fig. 4d agree well with the experimental observations along the stacking direction, namely, the $[103]_m$ zone axis (Supplementary Fig. S15).

Fig. 5. Structural projection of vaterite along $[130]_m$ direction. (a) SAED patterns. (b) Atomic resolution TEM image. (c) Atomic resolution HAADF image.

After tilting the grain by an additional 30° along the $(001)_m^*$ direction, the resulting $[130]_m$ zone-axis is obtained. As depicted in Fig. 5a, the SAED patterns indicate that the weak spots and diffused streaking feature between $(001)_m$ and $(002)_m$, which were observed in EDPs along the

$[010]_m$ direction (Fig. 4a), are almost absent. Additionally, the atomic resolution TEM (Fig. 5b) and HAADF images (Fig. 5c) do not exhibit any evident stacking disorders. This can also be rationalized by the concept of orientation variants, where tilting variant I to the $[130]_m$ direction results in variants II and III being tilted to $[100]_m$ and $[1\bar{3}0]_m$, respectively. The simulated EDPs along these three orientations are identical (Supplementary Fig. S16). By overlapping the EDPs along these three orientations, the experimental observations shown in Fig. 5a can be reproduced.

Second-order phase transition in vaterite

After clarifying the stacking principles of carbonate layers along the z -axis (Supplementary Note VII and Fig. S14), the question remains as to how the carbonate groups align in a single layer. Previous vaterite structure models (29, 34), including the $C2$ structure, show that the carbonate groups tilt slightly away from their high-symmetry orientations. Thus, to unveil the real orientations of the carbonate groups in vaterite, we conducted a detailed analysis of their atomic positions, using the $C2$ phase as the initial structure. We observed a small rotation of carbonate as the temperature increases, with the $C2$ phase transforming into the $C2/c$ phase at approximately 190 K, resulting in a higher symmetry. To quantify this phenomenon, we introduced an order parameter, Q , which measures the difference in the coordinates of the oxygen atoms in adjacent carbonate groups (see methods and Supplementary Note VIII). As shown in Fig. 6a, by periodically increasing/decreasing the temperature of the system, we find that Q fluctuates around zero at high temperature, corresponding to a high-symmetry (HS) state (i.e., $C2/c$), and deviates away from zero at low temperatures, corresponding to a low-symmetry (LS) state (i.e., $C2$). Figure 6b illustrates the orientations of carbonate groups in the HS and LS states. It is worth noting that, the other low-symmetry structure Cc was not observed in our simulations, as it would transform into the $C2$ structure at a relatively low temperature due to its lower stability.

Fig. 6. Second-order phase transition in vaterite. (a) Temperature and order parameter Q as a function of simulation time. (b) Schematic illustration of the carbonate group orientations in high-symmetry (HS) and low-symmetry (LS) states. (c) Two-dimensional free energy surface of vaterite as functions of order parameter Q and temperature. (d) Landau parameters as a function of temperature. Inset: Ginzburg-Landau free energy surface below, at, and above the critical temperature T_c of vaterite.

To understand the kinetic correlation between the two states, the well-tempered metadynamics (49), a smoothly converging and tunable free-energy method, was employed with the order parameter Q serving as the collective variable. Figure 6c plots the free energy surface as a function of Q and temperature. At high temperatures, a parabola-like free energy surface is observed, while at low temperatures, an energy barrier separates two local minima with the opposite order parameter value. This energy barrier decreases continuously with increasing temperature and eventually disappears at the critical temperature T_c (190 ± 10 K), consistent with

the findings presented in Fig. 6a. At the same time, the order parameter decreases continuously to zero, indicating a second-order phase transition in the Ginzburg-Landau model (50).

We find the free energy can be well described by $F(T, Q) = aQ^2 + bQ^4$, where a and b are phenomenological quantities known as Landau parameters. As predicted by the Ginzburg-Landau model, a exhibits a linear temperature dependence near the critical point T_c , *i.e.*, $a(T) = \alpha(T - T_c)$, while parameter b is a constant. We estimated the Landau parameters by fitting the free energy surface in Fig. 6c and the results are shown in Fig. 6d. When the temperature is around T_c , both parameters can be well fitted. At relatively high temperatures, $a(T)$ starts to exhibit nonlinearity. Although $b(T)$ is not a perfect constant, the slope of its dependence on temperature is relatively small, and such characteristic has been reported in previous studies (51, 52). At T_c , the low energy area of the free energy surface becomes rather flat. We find that the LS structures with positive and negative Q values are mirror-symmetrized to each other and share the same space group $C2$. As demonstrated in Fig. 6a, the LS structures with different chirality can interconvert between each other passing through the HS structure. Such chirality feature in vaterite might be significant in some specific biosystems, in which one enantiomer might be favored (53).

Conclusions

In summary, a comprehensive investigation of biogenic vaterite from two different marine species, the lackluster pearls and the asteriscus of carp, was conducted by combining systematic TEM characterizations, crystallographic analyses, and advanced theoretical calculations. Eventually, the structural mystery of vaterite CaCO_3 is uncovered. The basic structure of vaterite  monoclinic lattice with a space group of $C2$ or $C2/c$, with lattice parameters of $a=12.3 \text{ \AA}$, $b=7.13 \text{ \AA}$, $c=9.4 \text{ \AA}$, and $\beta=115.480^\circ$. The difference between the $C2$ and $C2/c$ structures is characterized by the slightly different orientations of the carbonate groups within a single layer, which are sensitive to temperature, leading to different symmetries in vaterite. Importantly, the transformation between the low- and high-symmetry phases can be well described by a second-order phase transition with a critical point of approximately 190 K. Vaterite has a pseudo-hexagonal feature with a pseudo sixfold axis along the $[103]_m$ direction, and three variants that are formed due to a 60° rotation along this direction. In principle, vaterite is a polytypical structure formed by the nanoscale random intergrowth of the three orientation variants, which are interrelated along the $[103]_m$ direction. The stacking sequence “0” is absent, while “+” and “-” sequences are

randomly distributed along the $[103]_m$ direction based on the monoclinic lattice. Additionally, the localized structural and chemical features of minerals are correlated with environmental nucleation and growth conditions, such as temperature, pressure and presence of impurities. However, the intrinsic structural features of vaterite are similar and the configuration rules do not change. The atomic scale insights of vaterite obtained from lackluster pearls and the asteriscus in the otoliths of carp are generally applicable to all the vaterite samples from different origination.

Materials and Methods

Sample preparation and electron microscopy methods

The lackluster pearl (vaterite structure) and normal pearl (aragonite structure) were collected from Zhuji, Southeast China. The asteriscus otolith pairs were extracted from the carps originated at Baiyang lake in China. The sample was cut into different sections using a linear precision saw with a thickness of $\sim 500 \mu\text{m}$ and then ground to $\sim 100 \mu\text{m}$ using silicon carbide (SiC) papers. Thin foils with a diameter of $< 3 \text{ mm}$ were stuck on the molybdenum grid, then grounded using variant grit SiC papers, and polished with diamond paste to $\sim 10 \mu\text{m}$ and finally thinned by Ar ion milling in a Gatan precision ion polishing system (PIPS) using a low voltage (3-4 kV) to avoid possible ion beam damage. Furthermore, to improve the electronic conductivity, perforated samples were sputtered with carbon layer using a sputter machine (Quorum Q 150R ES) for $\sim 20 \text{ s}$.

Raman spectra were obtained in back-scattering mode at room temperature by utilizing the Jobin-Yvon HR800 micro-Raman system with a grating speed of 1800 *lines/mm*. This system was equipped with a liquid-nitrogen-cooled charge-coupled device (CCD) and a $\times 100$ objective lens (NA=0.9). The resolution of the Raman system was 0.35 cm^{-1} per pixel. Raman spectra were measured using a 532 nm laser. A tilt series of SAED patterns were recorded using a FEI Tecnai T12 BioTWIN at 120 kV with a LaB₆ thermionic gun. The tilting angles along the X/Y axis range from $-70^\circ/-35^\circ$ to $+70^\circ/+35^\circ$. Atomic scale resolution TEM and HAADF images were acquired using a JEM ARM200CF microscope which was operated at 200 kV . This microscope was equipped with a cold field-emission gun, a probe corrector, and the dual silicon drift detectors (SDDs). The point resolution in scanning TEM (STEM) mode is $\sim 78 \text{ pm}$. The convergence semi-angle for STEM imaging was $\sim 22 \text{ mrad}$. The inner and outer acceptance semi-angles for HAADF imaging were approximately 90 and 250 *mrاد*, respectively. Within this collection angle range, the intensity of images is dominated by incoherent *Z* contrast. The area of a single detector is 100 mm^2 and the solid angle of this energy dispersive spectrum (EDS) system is approximately 1.8 steradian (*sr*).

Simulations of the EDPs are performed using the CrystalDiffract software. Atomic resolution HAADF image simulations were performed using the Dr. Probe software (54). The simulation parameters were chosen close to the experimental conditions assuming that all aberrations have been corrected within the probe forming aperture corresponding to a 21 *mrاد* semi-convergence

angle. The annular detector collection angles range from 80 to 200 *mrad*. An effective Gaussian source profile of 0.1 *nm* full width at half maximum (FWHM) was utilized. Simulated images were obtained as series over sample thicknesses up to 10 *nm*.

Theoretical calculations

Ab initio molecular dynamics calculation

The *ab initio* molecular dynamics (AIMD) calculations were performed using the Vienna *ab initio* package (VASP) (55, 56). The general gradient approximation of Perdew-Burke-Ernzerhof (GGA-PBE) was adopted for the exchange-correlation functional (57, 58). The electron wave function was expanded using a plane wave with a cut-off energy of 400 *eV*. Reciprocal space integration was performed by setting the *k* spacing to 0.5 (59). The AIMD was carried out with an isothermal-isobaric (NPT) ensemble at the temperature of 1000 *K* using the Langevin thermostat (60) and at the pressure of one bar using the Parrinello-Rahman barostat (61, 62). The time step in the simulation was 1 *fs*.

Density functional theory (DFT) energy and force calculations

Single point energy and forces were calculated using VASP. The smearing parameter and exchange-correlation density functional were the same as those in the AIMD simulation, while the energy cut-off for this calculation was increased to 1000 *eV*.

Training deep neural network model

All training of calcium carbonate DNN models was performed with the package DeePMD-kit (45). The settings of the training parameters are as follows: the cut-off radius smoothly decays from 7.5 Å to 8 Å . The sizes of three hidden layers for the embedding network and three hidden layers for the fitting network were set to (20, 40, 80) and (240, 240, 240) respectively. The learning rate decays from 1.0×10^{-3} to 5.3×10^{-8} . The pre-factors of the energy, force and virial term in the loss function change from 0.04 to 0.2, 1000 to 1 and 0.04 to 0.2, respectively. The total number of training steps was set to 1×10^6 .

To establish a training set capable of covering the whole solidification process, first, we collected configurations from the AIMD trajectories, in which we fixed half of the calcium atoms in the supercell while carbonate ions were allowed to rotate freely at extremely high temperatures to obtain various liquid–solid interface structures. Approximately 13,000 configurations were obtained together with the corresponding energy, force, and virial data as the training set to establish the first-generation DNN model (DNN1). We next performed MD simulations on

Lammps (63) using DNN1 to enable adequate sampling of the configuration space at larger timescales. To make our DNN model accurately predict the energies and forces of the structures under various thermodynamic conditions, the MD simulations covered the temperature range 1-2000 K. Additionally, initial solid structures with different stacking sequences along the z -axis were also considered. Notably, the orientations of carbonates varied casually at high temperature before melting, which indicated that not only the structures with ordered carbonates distributed over three orientations in one layer, but also the structures with disordered carbonates were included in our training set.

Based on the configurations obtained in the previous steps, we performed DFT single point energy calculations and constructed a new training set to establish DNN2. We repeated the above steps to perform MD simulations under the corresponding thermodynamic conditions and calculated the single point energies for the configurations extracted from the trajectories. The energies and forces calculated by DFT were compared with the predicted values of DNN (Supplementary Fig. S13), and the configurations with large error were added to the training set for iterative optimization. The errors of the final DNN model in terms of the atomic energies and forces on the test sets were $1.81 \times 10^{-3} \text{ eV/atom}$ and 0.11 eV/\AA respectively.

Molecular dynamics simulations

The molecular dynamics simulations were carried out using Lammps, integrated with DeePMD-kit to utilize the DNN potential. The simulation system with 6480 atoms (1296 CaCO_3 units), consisting of 36 layers of calcium atoms and 36 layers of carbonates, was in an orthorhombic box with periodic boundary conditions in all three directions with the dimensions of $2.2017 \times 2.5406 \times 15.3321 \text{ nm}^3$.

To perform the following crystallization simulations, we first constructed a liquid–solid coexisting configuration in which the solid phase was composed of only one single layer of vaterite. This was done through a simulation at a high temperature of 1600 K using the NPT ensemble starting from a perfect structure of vaterite, in which one layer of calcium atoms was fixed. Thus, the ordered calcium carbonates transformed into disordered states except for the fixed layer of calcium atoms. Since calcium atoms exhibit a hexagonal crystal structure in vaterite, fixing a layer of calcium atoms would allow the system to grow more easily in the specific direction. Then, we used the obtained configuration to perform the crystallization simulations. The liquid–solid phase transition proceeded along the [001] direction of the hexagonal lattice during 1000 K relaxation of the NPT

mode. Our structure arises from the natural transformation kinetics and therefore indicates the intrinsic properties of the structure of vaterite along the [001] direction. After obtaining a complete solid phase, we further quench the system to 10 K in 1 ns to reduce the thermal fluctuations. Snapshots of the trajectory are presented in Extended Data Fig. 6. Several simulations under the same thermodynamic conditions, but with different random seeds, were performed to cross-validate the simulation results.

In the above simulations, temperature and pressure were controlled by the stochastic velocity rescaling thermostat (64) and the Parrinello-Rahman barostat, respectively. The timestep is set to 1 fs. The initial velocity of each atom was generated with the Boltzmann distribution. The linear momentum was rescaled by subtracting the momentum of the center of the mass every 1000 timesteps. Plumed (65) packaged in LAMMPS was used for performing well-tempered metadynamics (66) and analyzing the trajectories. Crystal structure visualization was achieved using Ovito (67) and Vesta (68).

Well-tempered metadynamics and free energy surface

To estimate the difference in free energy between the high symmetry (HS) phase and the low symmetry (LS) phase at different temperatures, we further performed well-tempered metadynamics (WTMetaD) with the order parameter Q introduced in Supplementary Note VIII. The bias potential of WTMetaD is a history-dependent function of the order parameter composed by the deposits of Gaussians, and the height of the additive Gaussian dwindles as the sampling proceeds. For the comprehensive descriptions please refer to (68). In this work, the width and the initial height of Gaussian are set to 0.075 Å and 3 kJ/mol, respectively. The bias factor, which characterizes the rate of change of the deposited Gaussian height, is set to 30. The time interval of the deposits of Gaussians is equal to 500 fs. We calculated the free energy surfaces at 24 different temperatures with a temperature interval of 10 K from 50 K to 250 K. The other MD settings for the simulations were the same as those used in the growth process as described above. The simulation time was set as 2×10^6 steps (2 ns) to reach convergence. After the sampling was finished, we reweighted the obtained data using an on-the-fly strategy (69) to evaluate the free energy surfaces.

References

1. R. E. Zeebe, J. C. Zachos, K. Caldeira, T. Tyrrell, Carbon emissions and acidification. *Science* **321**, 51-52 (2008).
2. J. MacAdam, S. A. Parsons, Calcium carbonate scale formation and control. *Rev. Environ. Sci. Biotechnol.* **3**, 159-169 (2004).
3. F. C. Meldrum, Calcium carbonate in biomineralisation and biomimetic chemistry. *Int. Mater. Rev.* **48**, 187-224 (2003).
4. Z. Zou *et al.*, A hydrated crystalline calcium carbonate phase: Calcium carbonate hemihydrate. *Science* **363**, 396-400 (2019).
5. B. V. Parakhonskiy, A. Haase, R. Antolini, Sub-micrometer vaterite containers: synthesis, substance loading, and release. *Angew. Chem. Int. Ed. Engl.* **51**, 1195-1197 (2012).
6. X. San *et al.*, Uncovering the crystal defects within aragonite CaCO₃. *Proc. Natl. Acad. Sci. U.S.A.* **119**, e2122218119 (2022).
7. Y. Y. Kim *et al.*, Tuning hardness in calcite by incorporation of amino acids. *Nat. Mater.* **15**, 903-910 (2016).
8. J. D. Rodriguez-Blanco, S. Shaw, L. G. Benning, The kinetics and mechanisms of amorphous calcium carbonate (ACC) crystallization to calcite, viavaterite. *Nanoscale* **3**, 265-271 (2011).
9. E. Seknazi, S. Mijowska, I. Polishchuk, B. Pokroy, Incorporation of organic and inorganic impurities into the lattice of metastable vaterite. *Inorg. Chem. Front.* **6**, 2696-2703 (2019).
10. P. J. M. Smeets, K. R. Cho, R. G. Kempen, N. A. Sommerdijk, J. J. De Yoreo, Calcium carbonate nucleation driven by ion binding in a biomimetic matrix revealed by *in situ* electron microscopy. *Nat. Mater.* **14**, 394-399 (2015).
11. P. J. M. Smeets *et al.*, A classical view on nonclassical nucleation. *Proc. Natl. Acad. Sci. U.S.A.* **114**, E7882-E7890 (2017).
12. A. F. Wallace *et al.*, Microscopic evidence for liquid-liquid separation in supersaturated CaCO₃ solutions. *Science* **341**, 885-889 (2013).
13. L. Qiao, Q.-L. Feng, Z. Li, Special vaterite found in freshwater lackluster pearls. *Cryst. Growth. Des.* **7**, 275-279 (2007).
14. T. Saito, E. Sakai, M. Morioka, N. Otsuki, Carbonation of γ -Ca₂SiO₄ and the mechanism of vaterite formation. *J. Adv. Concr. Technol.* **8**, 273-280 (2010).
15. J. D. C. McConnell, Vaterite from Ballycraigy, Larne, Northern Ireland. *Miner. Mag. J. Miner. Soc.* **32**, 535-544 (1960).
16. X. Song, Y. Cao, X. Bu, X. Luo, Porous vaterite and cubic calcite aggregated calcium carbonate obtained from steamed ammonia liquid waste for Cu²⁺ heavy metal ions removal by adsorption process. *Appl. Surf. Sci.* **536**, (2021).
17. K. G. M. D. Abeykoon, S. P. Dunuweera, D. N. D. Liyanage, R. M. G. Rajapakse, Removal of fluoride from aqueous solution by porous vaterite calcium carbonate nanoparticles. *Mater. Res. Express* **7**, 035009 (2020).
18. A. Obata, H. Ozasa, T. Kasuga, J. R. Jones, Cotton wool-like poly(lactic acid)/vaterite composite scaffolds releasing soluble silica for bone tissue engineering. *J. Mater. Sci. Mater. Med.* **24**, 1649-1658 (2013).
19. Q. Huang, Y. Liu, Z. Ouyang, Q. Feng, Comparing the regeneration potential between PLLA/Aragonite and PLLA/Vaterite pearl composite scaffolds in rabbit radius segmental bone defects. *Bioact. Mater.* **5**, 980-989 (2020).
20. D. B. Trushina, T. V. Bukreeva, M. V. Kovalchuk, M. N. Antipina, CaCO₃ vaterite microparticles for biomedical and personal care applications. *Mater. Sci. Eng. C Mater. Biol. Appl.* **45**, 644-658 (2014).

21. D. Gebauer, A. Völkel, H. Cölfen, Stable prenucleation calcium carbonate clusters. *Science* **322**, 1819-1822 (2008).
22. K. Henzler *et al.*, Supersaturated calcium carbonate solutions are classical. *Sci. Adv.* **4**, eaa06283 (2018).
23. W. Sun, S. Jayaraman, W. Chen, K. A. Persson, G. Ceder, Nucleation of metastable aragonite CaCO₃ in seawater. *Proc. Nat. Acad. Sci. U.S.A.* **112**, 3199-3204 (2015).
24. D. B. Trushina, T. N. Borodina, S. Belyakov, M. N. Antipina, Calcium carbonate vaterite particles for drug delivery: Advances and challenges. *Mater. Today Adv.* **14**, (2022).
25. L. H. Fu, C. Qi, Y. R. Hu, C. G. Mei, M. G. Ma, Cellulose/vaterite nanocomposites: Sonochemical synthesis, characterization, and their application in protein adsorption. *Mater. Sci. Eng. C Mater. Biol. Appl.* **96**, 426-435 (2019).
26. R. E. Noskov *et al.*, Golden vaterite as a mesoscopic metamaterial for biophotonic applications. *Adv. Mater.* **33**, e2008484 (2021).
27. G. Steciuk, L. Palatinus, J. Rohlicek, S. Ouhenia, D. Chateigner, Stacking sequence variations in vaterite resolved by precession electron diffraction tomography using a unified superspace model. *Sci. Rep.* **9**, 9156 (2019).
28. A. G. Christy, A review of the structures of vaterite: the impossible, the possible, and the likely. *Cryst. Growth. Des.* **17**, 3567-3578 (2017).
29. R. Demichelis, P. Raiteri, J. D. Gale, R. Dovesi, The multiple structures of vaterite. *Cryst. Growth. Des.* **13**, 2247-2251 (2013).
30. E. Mugnaioli *et al.*, *Ab initio* structure determination of vaterite by automated electron diffraction. *Angew. Chem. Int. Ed.* **51**, 7041-7045 (2012).
31. L. Kabalah-Amitai *et al.*, Vaterite crystals contain two interspersed crystal structures. *Science* **340**, 454-457 (2013).
32. H. J. Meyer, Struktur und fehlordnung des vaterits. *Z. Kristallogr.* **128** 183-212 (1969).
33. A. Le Bail, S. Ouhenia, D. Chateigner, Microtwinning hypothesis for a more ordered vaterite model. *Powder Diffr.* **26**, 16-21 (2012).
34. R. Demichelis, P. Raiteri, J. D. Gale, R. Dovesi, A new structural model for disorder in vaterite from first-principles calculations. *CrystEngComm* **14**, 44-47 (2012).
35. L. Qiao, Q. L. Feng, Study on twin stacking faults in vaterite tablets of freshwater lacklustre pearls. *J. Cryst. Growth* **304**, 253-256 (2007).
36. S. J. Pennycook, D. E. Jesson, High-resolution incoherent imaging of crystals. *Phys. Rev. Lett.* **64**, 938-941 (1990).
37. Y. I. Svenskaya *et al.*, Key parameters for size- and shape-controlled synthesis of vaterite particles. *Cryst. Growth. Des.* **18**, 331-337 (2017).
38. X. H. Guo, S. H. Yu, G. B. Cai, Crystallization in a mixture of solvents by using a crystal modifier: morphology control in the synthesis of highly monodisperse CaCO₃ microspheres. *Angew. Chem. Int. Ed. Engl.* **45**, 3977-3981 (2006).
39. T. Schuler, W. Tremel, Versatile wet-chemical synthesis of non-agglomerated CaCO₃ vaterite nanoparticles. *Chem. Commun.* **47**, 5208-5210 (2011).
40. G. Behrens, L. T. Kuhn, R. Ubig, A. H. Heuer, Raman spectra of vateritic calcium carbonate. *Spectrosc. Lett.* **28**, 983-995 (1995).
41. U. Wehrmeister, A. L. Soldati, D. E. Jacob, T. Häger, W. Hofmeister, Raman spectroscopy of synthetic, geological and biological vaterite: a Raman spectroscopic study. *Journal of Raman Spectroscopy* **41**, 193-201 (2010).
42. S. Kamhi, On the structure of vaterite CaCO₃. *Acta Cryst.* **16**, 770-772 (1963).
43. L. Dupont, F. Portemer, t. late Michel Figlarz, Synthesis and study of a well crystallized CaCO₃ vaterite showing a new habitus. *J. Mater. Chem.* **7**, 797-800 (1997).

44. J. Wang, U. Becker, Structure and carbonate orientation of vaterite (CaCO₃). *Am. Mineral.* **94**, 380 (2009).
45. H. Wang, L. Zhang, J. Han, W. E, DeePMD-kit: A deep learning package for many-body potential energy representation and molecular dynamics. *Comput. Phys. Commun.* **228**, 178-184 (2018).
46. H. Niu, L. Bonati, P. M. Piaggi, M. Parrinello, *Ab initio* phase diagram and nucleation of gallium. *Nat. Commun.* **11**, 2654 (2020).
47. M. Yang, T. Karmakar, M. Parrinello, Liquid-liquid critical point in phosphorus. *Phys. Rev. Lett.* **127**, 080603 (2021).
48. W. Jia *et al.*, Pushing the limit of molecular dynamics with *ab initio* accuracy to 100 million atoms with machine learning. SC20: International conference for high performance computing, networking, storage and analysis. IEEE Press, 1-14 (2020).
49. A. Laio, M. Parrinello, Escaping free-energy minima. *Proc. Nat. Acad. Sci. U.S.A.* **99**, 12562-12566 (2002).
50. Y.-S. Ho, Review of second-order models for adsorption systems. *J. Hazard. Mater.* **136**, 681-689 (2006).
51. S. Radescu, I. Etxebarria, J. M. Perez-Mato, The Landau free energy of the three-dimensional Phi 4 model in wide temperature intervals. *J. Phys. Condens. Matter* **7**, 585 (1995).
52. M. Invernizzi, O. Valsson, M. Parrinello, Coarse graining from variationally enhanced sampling applied to the Ginzburg-Landau model. *Proc. Natl. Acad. Sci. U.S.A.* **114**, 3370-3374 (2017).
53. W. Jiang *et al.*, Chiral acidic amino acids induce chiral hierarchical structure in calcium carbonate. *Nat. Commun.* **8**, 15066 (2017).
54. J. Barthel, Dr. Probe: A software for high-resolution STEM image simulation. *Ultramicroscopy* **193**, 1-11 (2018).
55. G. Kresse, J. Hafner, *Ab initio* molecular dynamics for liquid metals. *Phys. Rev. B* **47**, 558-561 (1993).
56. G. Kresse, J. Furthmüller, Efficient iterative schemes for *ab initio* total-energy calculations using a plane-wave basis set. *Phys. Rev. B* **54**, 11169-11186 (1996).
57. P. E. Blöchl, Projector augmented-wave method. *Phys. Rev. B* **50**, 17953-17979 (1994).
58. G. Kresse, D. Joubert, From ultrasoft pseudopotentials to the projector augmented-wave method. *Phys. Rev. B* **59**, 1758-1775 (1999).
59. H. J. Monkhorst, J. D. Pack, Special points for Brillouin-zone integrations. *Phys. Rev. B* **13**, 5188-5192 (1976).
60. M. P. Allen, J. Tildesley D, *Computer simulation of liquids*. (Oxford University Press, 2017).
61. M. Parrinello, A. Rahman, Crystal structure and pair potentials: A molecular-dynamics study. *Phys. Rev. Lett.* **45**, 1196-1199 (1980).
62. M. Parrinello, A. Rahman, Polymorphic transitions in single crystals: A new molecular dynamics method. *J. Appl. Phys.* **52**, 7182-7190 (1981).
63. A. P. Thompson *et al.*, LAMMPS - a flexible simulation tool for particle-based materials modeling at the atomic, meso, and continuum scales. *Comput. Phys. Commun.* **271**, 108171 (2022).
64. G. Bussi, D. Donadio, M. Parrinello, Canonical sampling through velocity rescaling. **126**, 014101 (2007).
65. M. Bonomi *et al.*, Promoting transparency and reproducibility in enhanced molecular simulations. *Nat. Methods* **16**, 670-673 (2019).
66. A. Barducci, G. Bussi, M. Parrinello, Well-tempered metadynamics: A smoothly converging and tunable free-energy method. *Phys. Rev. Lett.* **100**, 020603 (2008).
67. A. Stukowski, Visualization and analysis of atomistic simulation data with OVITO—the Open Visualization Tool. *Modell. Simul. Mater. Sci. Eng.* **18**, 015012 (2010).
68. K. Momma, F. Izumi, VESTA 3 for three-dimensional visualization of crystal, volumetric and morphology data. *J. Appl. Crystallogr.* **44**, 1272-1276 (2011).

69. G. Bussi, A. Laio, Using metadynamics to explore complex free-energy landscapes. *Nat. Rev. Phys.* **2**, 200-212 (2020).

Acknowledgements: The authors are grateful to Prof. Qingling Feng at Tsinghua University for kindly providing the bulk pearl samples and Dr. Chi Zhang in Ocean University of China for many helpful discussions on vaterite structures.

Funding: X.S. acknowledges the National Natural Science Foundation of China (No. 51901065), the Nature Science Foundation of Hebei Province (No. E2020201023), and the Advanced Talents Incubation Program of Hebei University (No. 521000981164) for financial support. H.N. was supported by the National Science Fund for Excellent Young Scientist Fund Program (Overseas) of China, the Science and Technology Activities fund for Overseas Researchers of Shanxi Province, and the Research Fund of the State Key Laboratory of Solidification Proceeding (NPU) of China (No. 2020-QZ-03). This work made use of the EPIC facility of Northwestern University's NUANCE Center, which has received support from the SHyNE Resource (NSF ECCS-2025633), the IIN, and Northwestern's MRSEC program (NSF DMR-1720139).

Contributions: X.S. and X.H. performed the microscopy experiments and data analysis. J.H., M.C. and H.N. performed the theoretical calculations. X.H. and H.N. conceived and coordinated the entire project. All authors contributed to the discussion and writing of the manuscript.

Competing interests: The authors declare that they have no competing interests.

Data and materials availability: All data needed to evaluate the conclusions of this work are presented in the paper and the Supplementary Materials. Additional data related to this paper may be requested from the authors.

Extended Data Fig. 1. Microstructure and chemical features of aragonite. (a) Raman spectra of the pearls with aragonite and vaterite structures. The inset microscopic image shows the general appearance of the lackluster and normal pearls. (b) XRD patterns of the freshwater lackluster pearl. Bottom vertical pink lines show the peak positions of vaterite structure from PDF-33-0628. (c-e) C, Ca and O maps of vaterite, corresponding to the region shown in Fig. 1b. The inset in (d) is the EDS spectrum of vaterite.

Extended Data Fig. 2. Pseudo-hexagonal symmetry within vaterite. (a) Convergent beam electron diffraction patterns of vaterite along the same direction axis as Fig. 2b. (b) Duplicate EDPs of Fig. 2b. Red circles indicate some of the Set-I patterns with higher intensity. Green circles indicate some of the Set-II patterns with relatively lower intensity. (c) Atomic resolution TEM images of vaterite along the $[103]_m$ direction. (d) Digital fast Fourier transform (FFT) patterns of the HRTEM image in (c).

Extended Data Fig. 3. Stacking features of vaterite. (a) Structural projection of the monoclinic $C2$ phase along the $[010]$ direction. The shot purple lines indicate the stacking features of carbonate layers. The unit cell is denoted by the solid black lines, while the dashed box encloses a hexagonal lattice description of the $C2$ structure. (b) Stereo schematic showing the general arrangements of calcium atoms and carbonates in vaterite. One of the C-O bonds should point towards the edge of the trigonal prism, resulting in three possible orientations of carbonates that occupy half of the Ca_6 trigonal prismatic interstices. (c) The possible stacking sequences composed of three adjacent layers, which can be marked as “+”, “-”, and “0”, respectively.

Extended Data Fig. 4. Polytypic features within vaterite. Schematic illustration showing the formation of the orientation variant. (a) Pristine structure with uniform orientations. (b) Polytypic structure with variants. Variants II and III can be obtained by continuous 60° rotation of the carbonate layers from variant I along the $[103]_m$ direction. The dark area in (b) indicates the disordered stacking between different variants.

Extended Data Fig. 5. Microstructural features for the asteriscus pairs of carp having vaterite structure. (a) Raman spectrum. Inset is the optical image. (b) Low magnification TEM image. (c) HRTEM image and inset corresponding FFT patterns along $[010]_m$ zone axis again showing the representative faulted feature in vaterite. (d) Atomic resolution HAADF image along the $[010]_m$ zone axis showing the polytypic features of vaterite. The lines indicate the stacking feature of the Ca-C-O chains along the $[001]_m$ direction.

Extended Data Fig. 6. Crystallization simulation of vaterite growth from liquid. Snapshots of molecular dynamics simulations of vaterite growing along the z -axis direction during the first 9 ns at 1000 K after fixing a layer of calcium atoms and quenching to 10 K during the last 1 ns . The simulation cell contains 6480 atoms (1296 CaCO_3 units). The green atoms in the snapshot of 0 ns indicate the fixed layer of calcium atoms. The pink, blue, and yellow–green spheres represent calcium, carbon, and oxygen atoms, respectively.

Response to reviewers' comments

Reviewer #1 (Remarks to the Author):

Dear Authors,

This work is very interesting. It is intended to contribute to complete the structural and phase transformation descriptions of vaterite, this latter having been matter of large debates over many decades.

Response: We thank Prof. Chateigner for your time in reviewing our work and providing us with the constructive comments. We have addressed your following comments point-by-point.

I think that I am missing few experimental evidence in order to be able to accept your views concerning disorders in vaterite, which prevent me from giving out a clean opinion. All vaterite samples I could see up to now, biogenic or pure syntheses, exhibit diffraction signatures with extra peaks out of any "regular" modeling. Regular means here using classical 3D space groups, including faulted samples. This has been the purpose since the beginning of this long debate. In your TEM measurements you are indicating "diffuse" scattering between main peaks, the ones that you faulted model are aiming at reproducing. However, in the shown TEM diffraction patterns (Fig 3) such signatures are not so much visible, at least not enough to see if these "diffuse signals" are indeed diffuse or peaky, eventually coming from aperiodic peaks, commensurate or not. I mean, is your approach able to reproduce such signatures correctly or not, using usual space group + disorders and faults? Unfortunately, the promised XRD diagram is not provided (or I have not been able to find it).

Response: We are sorry for any confusion in our previous manuscript. Yes, our approach is able to reproduce all the experimental signatures correctly. First, we want to clarify that the diffuse features in Figure 3 are different from the diffuse signals in diffraction. The diffuse features in Figure 3 come from the CO₃ groups. Due to the pseudo hexagonal features of the C2 lattice, structural projections of the C2 lattice along [110]_m, [1-10]_m, [010]_m directions are very similar particularly the arrangement of Ca layers. The only difference among these three projections is the CO₃ configurations. Considering that both C and O are very light in contrast to Ca, their contrast in HAADF imaging is very weak. However, based on the HAADF imaging simulations, we can still see the diffuse features which is resulted from the CO₃ configurations. These diffuse features in HAADF imaging can be used to differentiate the orientation difference in the atomic resolution HAADF imaging as shown in Figure 4b.

For the diffuse signals that you refer to, we can observe them in Figure 2 and Figure 4a. For vaterite structure, many space groups are proposed to interpret it. But none of them can be used to index our serial EDPs. However, the one with monoclinic lattice (C2 or C2/c, a=12.2 Å, b=7.2 Å, c= 9.3 Å and β=115.2°) can be used to index most of the EDPs shown in Figure 2 except the red circle indicated spots (extra peaks out of any regular modeling) in Figure 2a. Meanwhile, based on the EDPs shown in Figure 4a, we find that vaterite is heavily faulted. Thus, we propose that vaterite should possess the polytypic features and the basic

structure has the C2 lattice. Due to the pseudo hexagonal features of the C2 lattice, 60° rotation domain is very likely to exist, which is also confirmed both experimentally (Figure 4b) and theoretically (Figure 4d). Based on the concept of polytypic features with stacking faults and three orientation domains, we did the electron diffraction simulations, which perfectly reproduce all the extra weak spots in Figure 2a, which is given in Supplementary Figure S20 as shown below.

Regarding the XRD diagram that possesses signatures with extra peaks, we will provide it and offer explanations in the subsequent sections.

Fig. S20. (a) Local magnification of Fig. 2a EDPs. (b) Simulated EDPs of the theoretically grown polytypic structure along the stacking direction.

On the other hand, the work by Steciuk et al. (ref 27 in the main article, or 11 in the supplements) clearly is able to fit all these extra peaks, observed also in the precession images, using aperiodic space group formalisms. Some of your HRTEM images clearly show atomic modulations that very probably could be understood using such 3D aperiodicities, in space groups larger than 3-dimensional.

Then the main issue I would like to be fixed before considering acceptance of this work would be to see on XRD and/or TEM diffraction patterns if the proposed models are able to reproduce the extra peaks observed by so many authors and which created the vaterite conundrum! If this is the case, then we could very interestingly compare your approach with superspace descriptions.

I think you haven't read enough carefully Steciuk's work: it is not concentrated on the C2/c space group, but on its aperiodic variants in superspace able to reproduce the extra peaks. This is why I insist on this. Please provide 1D fitted patterns in which we see the intensities reproduction using your faulted models.

Response: Thank you for mentioning Dr. Steciuk's work, which is a very valuable work to show that the structure of vaterite can be described using a high dimensional space group.

Researchers gradually recognize that the structure of vaterite is complicated and cannot be described based on any proposed pure lattice structure. In our work, we are attempting to resolve the structure of vaterite by using the direct atomic resolution imaging analysis and machine learning assisted theoretical calculations. As illustrated in Supplementary Figure S15 shown below, vaterite structure is heavily faulted. By using the

concept of polytypism with both stacking faults and orientation domains, we successfully rationalize all of our experimental data including the large angle tilted serial EDPs and atomic resolution HAADF images. More importantly, by using the machine learning assisted theoretical calculations, we directly grow the structure from atmosphere state into the crystalline vaterite structure with polytypic features. The theoretically grown structure verifies the fact that vaterite is indeed a polytypic structure with many stacking faults and orientation domains which is extracted based on our experimental observations. The electron diffraction simulations of the theoretically grown structure perfectly reproduced the extra spots observed in our EDPs as shown in Supplementary Figure S20.

Fig. S15. Atomic resolution HAADF image along the $[010]_m$ direction showing the polytypic features within vaterite along the $[001]_m$ direction. The red vertical lines indicate the stacking feature of the Ca-C-O chains along the $[001]_m$ direction. The inserted symbols indicate the stacking sequences of the carbonate layers introduced in Fig. S8.

To address this concern, we have further performed the Rietveld refinement analysis of the powder XRD data against the C2 model and the calculated supercell model introduced in Fig. 4d. The refinement results

are shown in Fig. 6c and 6d. For C2 model, the agreement factor is around 21.88%. By introducing the stacking faults and orientation domains, the agreement factor is significantly improved to 12.07%. It should be noted that the supercell model used to calculate the theoretical XRD pattern is just one representative configuration obtained from a single molecular dynamics' simulation. In other words, this supercell model does not encompass all possible stacking sequences of the bulk materials. However, it still exhibits a reasonable match with the XRD data. For all refinements, we freely refined the background (six parameters), the unit cell parameters (according to symmetry), the zero offset, and the microstrain (one parameter). The atomic coordinates were fixed based on the initial values of the model and the thermal factors were set to be isotropic at $0.1 \times 10^3 \text{ \AA}^2$. Additionally, the instrumental parameters were determined by refining a standard LaB_6 sample (NIST 660c). Instrumental parameters were fixed for all three refinements. Furthermore, from Figure 6c and 6d, it is evident that our power XRD data accurately captures the presence of extra peaks, such as the ones with 2θ around 40° , which have been previously reported in other studies. Clearly, the calculated XRD pattern based on our model explains these extra peaks well.

Fig. 6. Polytypic features within vaterite and XRD Rietveld refinement. Schematic illustration showing the formation of the orientation variant. (a) Pristine structure with uniform orientations. (b) Polytypic structure with variants. Variants II and III can be obtained by continuous 60° rotation of the carbonate layers from variant I along the $[103]_m$ direction. The dark area in (b) indicates the disordered stacking between different variants. Rietveld refinement of the experimental powder XRD pattern collected at room temperature against the (c) C2 model and (d) proposed polytypic model that contains stacking faults and orientation domains. Experimental data points are shown in cross-symbols, the calculated powder pattern is shown in green, the background is shown in red, and the difference curve between the experimental and calculated data is shown in blue color.

Last comment: using simulations, you justify the absence of so-called "0" variants on the basis of a 41 meV/f.u. difference. I was wondering if such a "small" magnitude is reachable using the simulations used, and if yes, what would be the associated uncertainties? Aragonite is at +1 kJ/mol from calcite in normal conditions, and vaterite at much more, then presumably 41 meV/f.u. can be considered small enough not to consider full absence of "0" variants?

Response: Firstly, the root means square error of the potential used in our simulations. It is estimated to be 1.81 meV/atom (see Materials and Methods and supporting material Note VII). This level of precision is fully capable of capturing the energy differences observed in the simulations. Additionally, it is worth noting that an energy difference of 41 meV/f.u. unit is equivalent to 3.96 kJ/mol, surpassing the energy gap between vaterite and calcite (2.9 kJ/mol). Hence, it is reasonable that the "0" arrangement does not appear in either our experimental or theoretical results. While we cannot completely dismiss the possibility of "0" variants forming under specific conditions, the probability of their occurrence is significantly smaller compared to the other two variants (+ and -).

This work is potentially of large impact, but I really need to be able to see how the proposed faults are ameliorating diffraction fits.

Response: We thank Prof. Chateigner for your constructive comments. Dr. Malliakas is a XRD expert in our university. He helped us collect more XRD data and performed detailed Rietveld refinement analysis against different models. It is found that XRD data cannot be indexed well by any unknown pure lattice. However, by introducing the stacking faults and orientation domains into the C2 lattice, the agreement factor is significantly improved. Most importantly, our model is capable of describing all the experimental signals, including the extra peaks, well. We hope your previous comments have been well addressed by these detailed responses. If you have further questions, we are very happy to have further discussions with you.

Reviewer #2 (Remarks to the Author):

It is a solid piece of work on a very important type of material, using advanced imaging methods that are supported by MD simulations. Machine learning was applied as well. The Supplementary Notes are really good and they make a fascinating read, showing at the same time, the depth that authors have gone to trying to understand the challenges of defining a crystal structure of vaterite.

Response: Thank the reviewer for the affirmative and constructive comments on our work. We have addressed your following comments point-by-point.

I do, however, have issues with the way the results are presented and more importantly not extending them to the results obtained by other methods. Synchrotron x-ray and neutron diffraction are the methods specifically developed for solving complex crystal and magnetic (in the case of neutrons) structures. Both are volume-averaged techniques that can help to understand to what extent the findings about stacking faults obtained with a local probe such as HRTEM and HAADF are applicable to the entire sample. In all the applications that the authors mentioned in the introduction, a substantial volume of vaterite will be used. How do your results map on the samples of several cubic cm? Neutrons are sensitive to light elements like O and C, providing a much better contrast than lab XRD. The authors had a discussion in the SI about difficulties to understand lab XRD results and briefly dismissed synchrotron x-rays results as well. It is puzzling because synchrotron x-rays provide superior resolution, small background. Using a 2D position-sensitive detector can show you presence of the crystallites in the powdered samples. X-ray diffraction measurements on a single nanoparticle are now possible, not to mention anomalous x-ray scattering can be done as well. Yet, all the scattering results are dismissed based on lab XRD work.

I would like to see an extended discussion on why neutron and scattering methods could not be applied to vaterite. There are many good examples of how microscopy and scattering results complement each other (<https://doi.org/10.1038/ncomms6964>, [10.1039/D0NR08615K](https://doi.org/10.1039/D0NR08615K), just to name a few). Did the authors do the Rietveld analysis of their XRD data presented in Ext. Fig. 1b? Did you try to simulate x-ray and neutron diffraction patterns can compare them with the literature? This is a necessary piece, which is missing.

Response: Thank the reviewer for the comments on synchrotron X-ray and neutron diffraction techniques. Indeed, these techniques are very powerful toolsets for solving complex crystal structures. Researchers have also been tried to resolve the structure of vaterite using synchrotron X-ray and neutron diffraction (synchrotron x-ray: *Angewandte Chemie International Edition*, 2012, 51(28): 7041-7045; *Journal of Crystal Growth*, 2014, 407: 78-86; *Scientific reports*, 2019, 9(1): 9156; neutron diffraction: *Scientific Reports*, 2016, 6(1): 36799.). But the derived conclusions are still controversial. Based on the neutron diffraction data, Chakoumakos *et al* (*Scientific Reports*, 2016, 6(1): 36799) indicated that the hexagonal structural model with space group $P6_522$ provides the best fit, which agrees well with the model provided based on synchrotron X-ray diffraction data (*Journal of Crystal Growth*, 2014, 407: 78-86). However, based on the synchrotron X-ray data, Mugnaioli *et al* (*Angewandte Chemie International Edition*, 2012, 51(28): 7041-7045) proposed that vaterite structure likely could be labeled by a monoclinic structure ($a=1.217$ nm, $b=0.712$ nm, $c=2.532$ nm, $\beta =99.228^\circ$) and Steciuk *et al* however (*Scientific reports*, 2019, 9(1):9156) proposed that vaterite should be labeled by a superspace group.

Previous results already demonstrated that synchrotron X-ray and neutron diffraction techniques are very limited here and cannot provide exclusive conclusions for vaterite structure. The main reason is that the structure of vaterite is heavily faulted as shown in Supplementary Fig. S15 shown below. We propose that vaterite should be regarded as polytypic structure, in which the stacking is random along $[103]_m$ direction. There are not only many stacking disorders but also three orientation domains. Within even 1-2 nm, stacking disorders and/or stacking faults occurs. The orientation domain which was resulted from localized stacking orders is also very small (< 3 nm) as shown in Figure 4b. The spatial resolution of synchrotron X-ray and neutron diffraction is around 10-30 nm and 20 μm , respectively. Within this scale, there are always lots of stacking disorders in vaterite. Thus, synchrotron X-ray and neutron diffraction techniques can only provide an averaged structural information, where many nanoscale features including stacking disorder and domains are averaging out. In summary, as for vaterite, we believe that the single variant size is too small and the density of stacking faults are too high that it is very challenging to use the X-ray diffraction and neutron diffraction to resolve its structure, particularly the atomic modulations.

Fig. S15. Atomic resolution HAADF image along the $[010]_m$ direction showing the polytypic features within vaterite along the $[001]_m$ direction. The red vertical lines indicate the stacking feature of the Ca-C-

O chains along the [001]_m direction. The inserted symbols indicate the stacking sequences of the carbonate layers introduced in Fig. S8.

However, based on the commercial usage of the aberration corrector, the spatial resolution of TEM is as high as 0.07 nm, which is a perfect toolset to resolve localized structure order/disorder. Thus, we are able to resolve the polytypic features of vaterite at the atomic scale, as shown in Supplementary Fig. S15. Additionally, based on the polytypic model we proposed, which is also again conformed by machine learning assisted theoretical calculations. The electron diffraction simulations of the theoretically grown structure perfectly reproduced the extra spots observed in our EDPs as shown Supplementary Figure S20.

In addition, we performed the Rietveld refinement analysis of the powder XRD data against the C2 model and calculated supercell model introduced in Fig. 4d. The refinement results are shown in Fig. 6c and 6d. For C2 model, agreement factor is around 21.88%. By introducing the stacking faults and orientation domains, the agreement factor is significantly improved to 12.07%. It should be noted that the supercell model used to calculate the theoretical XRD pattern is just one representative configuration obtained from a single molecular dynamics' simulation. In other words, this supercell model does not encompass all possible stacking sequences of the bulk materials. However, it still exhibits a reasonable match with the XRD data. For all refinements, we freely refined the background (six parameters), the unit cell parameters (according to symmetry), the zero offset, and the microstrain (one parameter). The atomic coordinates were fixed based on the initial values of the model and the thermal factors were set to be isotropic at $0.1 \times 10^3 \text{ \AA}^2$. Additionally, the instrumental parameters were determined by refining a standard LaB₆ sample (NIST 660c). Instrumental parameters were fixed for all three refinements.

Fig. S20. (a) Local magnification of Fig. 2a EDPs. (b) Simulated EDPs of the theoretically grown polytypic structure along the stacking direction.

Fig. 6. Polytypic features within vaterite and XRD Rietveld refinement. Schematic illustration showing the formation of the orientation variant. (a) Pristine structure with uniform orientations. (b) Polytypic structure with variants. Variants II and III can be obtained by continuous 60° rotation of the carbonate layers from variant I along the $[103]_m$ direction. The dark area in (b) indicates the disordered stacking between different variants. Rietveld refinement of the experimental powder XRD pattern collected at room temperature against the (c) C2 model and (d) proposed polytypic model that contains stacking faults and orientation domains. Experimental data points are shown in cross-symbols, the calculated powder pattern is shown in green, the background is shown in red, and the difference curve between the experimental and calculated data is shown in blue color.

Fig. S22. (a) Comparison of the experimental powder diffraction data against a Silicon standard (NIST 640d) and (b) Zoomed in region of left figure. The reflections from Si are sharper than those from the sample indicating that the instrumental resolution is sufficient for probing broadening due to defects in the sample.

We ensured that the resolution of the powder diffractometer used for this experiment was high enough to resolve broadening of reflections due to the defects. Figure S22 shows the experimental powder data of the sample against a Silicon standard (NIST 640d) which contains no strain/defects. Instrumental resolution is narrower than the resolution needed to resolve the broad peaks observed for the sample. We believe the superior resolution from a synchrotron source will not provide additional unique information given that the broadening of the peaks for this particular sample is intrinsically large. Additionally, we performed simulations of powder patterns using the proposed model for X-rays and neutrons for the same wavelength of 1.54187 Å. As shown in Figures S23 and S24, it is apparent that neither X-rays nor neutrons give significant contrast in the entire q -range. Certain regions appear better for X-rays and other regions appear better for neutrons. As recommended by the reviewer, we have included additional discussion as Supplementary Note VIII in the revised manuscript.

Fig. S23. (a) Comparison of the simulated powder diffraction of the proposed model using X-ray and neutron of the same wavelength. (b) Zoomed in region of left figure. Neither X-rays nor neutrons appear to give significant contrast in the entire q-range.

Fig. S24. (a) Fig. S23 in logarithmic scale and (b) Zoomed in region of left figure.

More specific comments:

p.1, Abstract, first sentence: "... plays a critical role in various fields". What fields? Fields of applications or research?

Response: Thanks to the reviewer for the valuable comment. We have added some examples such as hydrosphere, biosphere, and climate regulation.

p.3: Why these two particular samples were chosen for this work? What is so special about them? Please, elaborate on this topic the Intro section.

Response: Thanks to the reviewer for the valuable comment. Vaterite is an unstable crystalline form of CaCO_3 and crystallinity is usually not good. The lackluster pearl is a representative sample having good crystallinity, which is very suitable for high quality atomic resolution imaging analysis. That is why we mainly focus the deep analysis based on this sample. But to verify that our extracted information is applicable to other vaterite samples, we performed detailed observations on another vaterite sample with relatively good crystallinity, namely the asteriscus otolith pairs. We find that the polytypic features observed in lackluster pearl also exist in the asteriscus otolith pairs. Thus, we believe that the polytypic features should be general to all vaterite samples. More supporting evidence is listed in the following points.

p.9: Why do you claim that the obtained structural information is intrinsic to all vaterite samples? Is this conclusion based on an analysis of just two samples? Please, elaborate more. The same sentence is repapered in the Conclusions.

Response: Vaterite is a widely existing material that is distributed over the earth in materials ranging from sedimentary rocks to biological hard tissues. All vaterite samples exhibit diffraction signatures with extra peaks out of any "regular" modeling, whether it's biogenic or pure syntheses (D. Chateigner). This is because there are huge planar defects (twins/stacking faults) in the vaterite samples (biogenic sample-science, 2013, 340(6131): 454-457, Journal of crystal growth, 2007, 304(1): 253-256, syntheses sample-Angewandte Chemie International Edition, 2012, 51(28): 7041-7045). Based on different kinds of analysis techniques, the obtained structural information in our work is intrinsic for all kinds of vaterite samples. More supporting evidence are listed below.

1> XRD is a bulk measurement method, i.e., the spectral data reflects the averaged structural information of the samples. Therefore, we have performed additional XRD analysis. The XRD data of our lackluster pearls are shown in Figure X1a. As you can see, all the main peaks of our XRD data agree well with the standard diffraction data of vaterite listed in PDF-33-0628. Several XRD results of other samples containing a vaterite structure are shown in Figure X1b-d. Peak positions of our XRD results are in good agreement with these previous XRD results as well. This is clear evidence indicating that our lackluster pearls are representative biominerals containing a vaterite structure.

Figure X1. (a) XRD patterns of the vaterite sample in our study (freshwater pearl) and peak positions of vaterite from PDF-33-0628 shown in pink. (b) XRD patterns of the synthetic vaterite obtained from Reference (1). (c) XRD patterns of another synthetic vaterite from Reference (2). (d) XRD patterns of the synthetic vaterite from Reference (3).

2> Raman spectroscopy is another widely used bulk measurement technique. The Raman spectrum of our lackluster pearls are shown in Figure X2a. Raman spectra of synthetic samples with vaterite structure are displayed in Figure X2b-c. Our results are in good match with previous works. Additionally, the representative spectra feature of geological samples, synthetic samples and biogenic samples with vaterite structure are listed in Figure X2d. Our sample shows a typical triplet in the region between 1074 and 1091 cm^{-1} due to symmetric stretching of C-O bonds (ν_1), a representative feature of vaterite. This is additional evidence indicating that our lackluster sample can be categorized as a common biomineral containing a vaterite structure.

Table 4. Raman band positions (wavenumbers in cm^{-1}) and FWHM (cm^{-1}) of the internal modes of the carbonate ion in vaterite from literature and from this study

Vaterite	ν_4	ν_2	ν_1	ν_3
Synthetic vaterite ^[30]	667.2 (8.4), 674.0 (9.7) 684.8 (6.4), 738.4 (7.9) 743.5 (7.3), 751.3 (8.7)	873.7 (5.5) 877.6 (2.5) 881 (8.5)	1075.0 (7.0) 1081.4 (9.7) 1090.9 (6.6)	1421.1 (21.1), 1440.9 (16.7), 1459.9 (26.8), 1480.4 (31.4) 1542.3 (35.1), 1555.0 (5.6)
Biogenic vaterite from otolith of Coho Salmon (sample 1) ^[36]	685 (w), 740 (sh)	n.d.	1074 (s)	n.d.
Synth. vaterite sample 060829.05	744 (sh), 751 (w)		1081 (w) 1090 (vs)	1421 (14.3), 1440 (16.4)
Synth. vaterite sample 071217.04	666 (5.0), 672 (9.7) 685 (4.3), 738 (6.1) 743 (8.0), 751 (7.0)	873 (3.2), 877 (3.2)	1075 (5.6) 1081 (5.1) 1090 (5.3)	1456 (22.6), 1479 (27.0) 1557 1421 (14.8), 1441 (17.0)
Vaterite, geological sample No. 1	667 (6.2), 672 (8.0) 685 (6.5), 739 (8.0) 745 (8.4), 752 (7.4)	873 874 (9.0)	1075 (6.7) 1080 (5.9) 1091 (6.2) 1075 (4.8)	1460 (28.7), 1480 (30.9) 1554 1421 (12.4), 1438 (18.6)
Vaterite, geological sample No. 2	666 (5.1), 671 (4.3) 685 (3.7), 738 (4.8) 743 (7.0), 751 (6.5)	874 (9.0) 875 (10.7)	1075 (4.8) 1079 (5.0) 1090 (5.1) 1075 (4.9)	1461 (14.5), 1479 (22.8) 1555 1440
Vaterite, biogenic FWCP-China-176	666 (5.1), 671 (4.3) 685 (4.1), 738 (4.7) 743 (7.0), 751 (6.4)	874 (9.0) 873	1075 (4.8) 1081 (4.2) 1091 (4.5) 1074 (5.2)	1421 (12.4), 1438 (18.6) 1458 n.a.
Vaterite, biogenic FWCP-Biwa-11 spot 40	686 (4.5), 739 (4.9) 745 (8.4), 753 (6.5)	873	1074 (5.2) 1079 (5.4) 1090 (5.2) 1075 (4.7)	n.a. n.a.
Vaterite, biogenic FWCP-sample C spot 39	680 748	n.a.	1075 (4.7) 1079 (6.0) 1090 (5.1) 1075 (5.5)	n.a. n.a.
	670 740 751	n.a.	1075 (5.5) 1081 (4.8) 1090 (4.9)	n.a.

n.a., Not analyzed because of an overlap with pigment peaks^[31] or because of high luminescence.

Figure X2. (a) Raman patterns of the vaterite sample of our study (freshwater pearl). (b) Raman patterns of the synthetic vaterite Reference (3). (c) Raman patterns of the synthetic vaterite from Reference (2). (d) Raman patterns of the synthetic vaterite from Reference (4).

3> HRTEM imaging is a technique demonstrating the general structural features of materials locally at the nanometer scale. As shown in Figure X3, HRTEM images of our lackluster pearls and associated FFT

patterns are similar as those obtained from other samples with vaterite structure, characterized by their faulted structure. Such agreement strongly suggests that our lackluster pearls possess typical structural features observed in vaterite.

Figure X3. HRTEM micrograph of the vaterite sample in our study (freshwater pearl). (b) HRTEM micrograph of vaterite spicules of *H. momus* from Reference (5). (c) HRTEM micrograph of the synthetic vaterite from Reference (6). (d) HRTEM micrograph of the freshwater pearl from Reference (7).

- 4> Our work reveals that vaterite should be accurately regarded as a polytypic structure. The main structural features including the in-plane rotation and out-of-plane stacking of carbonates are uncovered for the first time by means of advanced experimental observations and theoretical calculations. For this polytypic structure, the atomic scale structure even varies slightly among different areas within one sample, however, the configuration rules of calcium and carbonates are fixed. Although the localized structure can be affected slightly by environmental conditions (as is also applicable to many other materials in general), the configuration rules of vaterite we discovered are essential here and generally apply to all other geological samples, synthetic samples, and biogenic samples.
- 5> The asteriscus of carp is indeed composed of vaterite, and the structural features at atomic scale are in excellent agreement with our results extracted based on the lackluster as shown in Fig.S16. This direct evidence verify that the uncovered main structural information extracted based on lackluster are generally applicable to other minerals having vaterite structure.

- 6> A hallmark of (bio)mineralization is that the mineral structure is dependent on environmental nucleation and growth conditions, such as temperature, pressure, presence of impurities, etc. Likewise, the vaterite structure may change slightly depending on such conditions. This is also demonstrated by the slight deviation in Raman spectra from abiotic vaterite. However, the intrinsic structural features of vaterite are the same and the configuration rules of the vaterite do not change. The main reason for selecting the lackluster pearl to uncover the structural mystery of vaterite is because their crystallinity is very good, which is convenient and essential for applications of various advanced characterization techniques. As shown in many publications (e.g., Kabalah-Amitai, et al, Science, 2013; Pokroy, et al, Nature Materials, 2004), researchers have to choose the representative samples for deep study but cannot enumerate the observations in every mineral with the same structure but different origination.
- 7> The initial stage of molecular dynamics simulation is based on calcium atoms configured in the hexagonal lattice that have been recognized by previous literature (8, 9). Additionally, the configuration rules revealed by the growth model of vaterite are accurate within first principles, which is in accordance with experimental results. The machine learning neural network model and the method of molecular dynamics are absolutely independent from environmental and other possible external conditions. We can safely conclude that the intrinsic characteristics of vaterite thus does not depend on the sources of the samples.

In conclusion, these evidences verify that the uncovered main structural information extracted based on lackluster is generally applicable to other minerals having vaterite structure. The results based on different sources of vaterite are the same is not surprising to us as the structural features we uncovered are the intrinsic information of vaterite.

p.11: Why the temperature-dependent studies important? What is your motivation? In the Intro section, most of the applications you mentioned operate at room temperature. Please, explain why temperature-induced phase transitions are important for vaterite.

Response: Temperature is an important environmental condition for nucleation, growth and phase transition of the vaterite. For chemistry, different temperature strongly affect the final reaction products and the microstructure of vaterite (Powder Technology, 2015, 284: 265-271, Powder Technology, 2022, 410: 117865.); For biology, temperature have an important effect on the crystallization process in the otoliths of the fish (Journal of Experimental Biology, 2017, 220(16): 2965-2969, Scientific Reports, 2021, 11(1): 13878.); For geochemistry, temperature is a key parameter for the kinetics and mechanism of the transformation of vaterite to calcite (Bulletin of the Chemical Society of Japan, 1973, 46(5): 1414-1417; Journal of Earth and Planetary Materials, 1971, 56(3-4_Part_1): 620-624; Journal of Thermal Analysis and

Calorimetry, 2000, 60: 463-472). Thus, the temperature is an important factor in tailoring the fine structural features of vaterite.

In addition, investigating the temperature-dependent behavior of vaterite at the atomic scale presents significant challenges. In this study, we have addressed this question by employing molecular dynamics simulations. Besides the stacking sequence of carbonate ions to understand the structure of vaterite, an additional important consideration is the symmetry associated with the slight distortion of the carbonate groups within a single layer, which shows temperature sensitivity. Our computational findings demonstrate that the vaterite phase corresponds to a high-symmetry structure at high temperatures, while adopting a low-symmetry structure at low temperatures. This discovery highlights the importance of understanding the temperature-dependent phase transition in order to comprehensively comprehend the structure of vaterite.

Reference

1. X.H. Guo, S.H. Yu, G.B. Cai, Crystallization in a mixture of solvents by using a crystal modifier: morphology control in the synthesis of highly monodisperse CaCO₃ microspheres. *Angew. Chem. Int. Ed. Engl.* 45:3977-3981 (2006).
2. Y.I. Svenskaya, et al., Key parameters for size- and shape-controlled synthesis of vaterite particles. *Cryst. Growth. Des.* 18:331-337 (2017).
3. T. Schuler, W. Tremel, Versatile wet-chemical synthesis of non-agglomerated CaCO₃ vaterite nanoparticles. *Chem. Commun.* 47:5208-5210 (2011).
4. Wehrmeister, U., Soldati, A. L., Jacob, D., Häger, T. & Hofmeister, W. Raman spectroscopy of synthetic, geological and biological vaterite: a Raman spectroscopic study. *Journal of Raman Spectroscopy: An International Journal for Original Work in all Aspects of Raman Spectroscopy, Including Higher Order Processes, and also Brillouin and Rayleigh Scattering* 41, 193-201 (2010)
5. L. Kabalah-Amitai, et al., Vaterite crystals contain two interspersed crystal structures. *Science* 340:454-457 (2013).
6. E. Mugnaioli, et al., Ab initio structure determination of vaterite by automated electron diffraction. *Angew. Chem. Int. Ed.* 51:7041-7045 (2012).
7. L. Qiao, Q.L. Feng, Study on twin stacking faults in vaterite tablets of freshwater lustrous pearls. *J. Cryst. Growth* 304:253-256 (2007).
8. A.G. Christy, A review of the structures of vaterite: the impossible, the possible, and the likely. *Cryst. Growth. Des.* 17:3567-3578 (2017).
9. R. Demichelis, P. Raiteri, J.D. Gale, R. Dovesi, The multiple structures of vaterite. *Cryst. Growth. Des.* 13:2247-2251 (2013).

Reviewer #3 (Remarks to the Author):

The paper presents evidence for vaterite being a polytype rather than a defined structure. This idea has been out for nearly a century, and proven and consolidated through both experiments and theory in the past decade:

Response: We appreciate the reviewer's time in looking at our work. The structure of vaterite has been debated for around one century until now, which means researchers are still not having enough accurate information to describe its real structural features. The concept of polytypic feature was new and proposed in the year of 2010 but not a century year ago (Wehrmeister et al. J. Raman Spectrosc. 2010, 41, 193). In the available literatures, most of the researchers tried to resolve the structural mystery by using bulk measurement techniques such as XRD and Raman. Meanwhile, many hypothesis (e.g., twinning, faults) have been proposed as well based on the bulk analysis. But until now, there is still even no atomic resolution images which can demonstrate the existence of these defects.

Indeed, there are several papers in in which high resolution transmission electron microscopy (HRTEM) images were included. But HRTEM imaging techniques is very limited for describing heavily faulted complex structures because it can only provide the phase contrast and the accompanying delocalization will introduce many artifacts. The atomic resolution high angle annular dark field (HAADF) imaging technique is the ideal method here to resolve the structural mystery vaterite. However, it is very challenging to obtain the atomic resolution HAADF images of vaterite. That is why there is no such kinds of data until now. As far as we know, for the first time, we not only directly observed the polytypic features by atomic resolution HAADF imaging but also rationalize the polytypic features by means of machine learning assisted theoretical calculations.

As the reviewer said that Meyer *et al* provided hypothesis for stacking faults by XRD in 1969, but Qiao *et al* observed twin stacking faults in 2007. From Meyer (Z. Kristallogr. 1969, 128, 183) to Christy (Cryst. Growth Des. 2017, 17, 6, 3567–3578), half a century has passed. Many scientists spent a lot of effort to investigate the structure of vaterite. The information of vaterite is getting more and more clear, but until now there are some fundamental problems unsolved. Qiao *et al* believed that vaterite is hexagonal, $P6_3/mmc$. No matter $P6_3/mmc$ or $P3_21$ or $C2$, there is still undetermined part for the vaterite structure. All of them cannot be used to indexed the bulk XRD data well.

- Meyer, Z. Kristallogr. 1969, 128, 183 (XRD): hypothesis for stacking faults

In this paper, hypothesis for stacking faults was first proposed. However, stacking faults and polytypic features are two different concepts. In generally, any planar defects can be described as stacking faults. But polytypic structure usually contains many ordered and disordered stackings. Stacking faults can broadly exist in many structures but the phase contain polytypic features are limited.

- Qiao & Feng, *J. Cryst. Growth*, 2007, 304, 253 (HRTEM & SAED): twin stacking faults observed. In this paper, the authors used HRTEM and SAED data to show the existence of twin and stacking faults. But the analysis is very basic since the HRTEM image intrinsically contain the phase delocalization and thus artifacts existed. Additionally, these data were obtained using the conventional TEM without imaging corrector. Thus, the spatial resolution is only around 2 Å and HRTEM image here cannot be used for direct correlation with atom configurations. Additionally, the proposed micro twin concept based on the HRTEM images is inaccurate because the features does not meet the definition of the twin. To descript a twin feature, at least five twin elements (K_1 , h_1 , K_2 , h_2 , twin disconnection and shear angles) are needed. But none of them are even mentioned in this paper.

- Wehrmeister et al. *J. Raman Spectrosc.* 2010, 41, 193 (Raman spectroscopy): hypothesis for polytypism, together with evidence for more than 2 independent CO₃²⁻ in the structure

In this paper, the authors proposed the concept of polytypism and the structure of vaterite contains at least three crystallographically independent carbonate groups and similar carbonate group layers. But there is no direct imaging evidence for this hypothesis. Meanwhile, the authors also stated that “vaterite may be not well ordered but is possibly affected by stacking faults, layer shifts or syntactic intergrowth irregularities, making it difficult to determine the crystal structure”. This is another evidence to show the complexity of vaterite structure, which deserves deep study.

- Demichelis et al. *CrystEngComm* 2012, 14, 44; *Cryst. Growth. Des.* 2013, 13, 2247 (DFT): re-analysing the whole set of experimental and theoretical structures, excluding the some of the proposed structures, finding others more compatible to experimental evidence and theoretical investigations, and providing evidence that different polytypes do not differ significantly in energy;

In this paper, the authors did detail DFT calculations to evaluate the structure of the vaterite with different structures. Based on the energetic considerations, the authors conclude that vaterite with space group P3₂21 is proposed as the best model for an ordered vaterite structure. However, the space group does not match our electron diffractions at all. Meanwhile, this DFT conclusion is based on the fact that vaterite is an order structure. But this is not accurate since previous literatures already show that many stacking faults are existed as well.

- Kabalah-Amitai et al. *Science* 2013, 340, 454 (HRTEM): at least 2 distinct structures for vaterite identified in *Herdmania momus* spicules, given as hexagonal and 1 undetermined; the main differences between the structures are related to CO₃²⁻ (as discussed in Wehrmeister et al 2010 and in Demichelis et al 2012, 2013 and in all the literature after then). Here it is true that the hexagonal space group selected does not explain any of the weak diffractions, however, this has been addressed in the literature from 2012 on;

In this *Science* paper, the authors tried to use the HRTEM data to analyze the structure. Although the HRTEM was obtained based on the aberration corrected TEM, the HRTEM technique is very limited in analyzing the complex structures with many faults since it can only provide the phase contrast and artifact

always exist. We cannot use HRTEM images for deep analysis since atom configurations cannot be directly correlated to the atom columns. Importantly, even the authors themselves claimed that “Vaterite consists predominantly of a hexagonal structure with at least one other coexisting crystallographic structure.”. But there is no discussion on these coexisting crystallographic features. That again confirms that many structural details of vaterite are still unknown.

-Balan et al. *Chemical Geology*, 2014, 374–375, 84 (QM calculation): mixing energies of sulfate into vaterite are consistent with one of the hexagonal models discussed above (one of the two polytypes, P3221); In this paper, the author again claimed that “A similar conclusion most likely holds for the P6₃/mmc model, which also differs from the minimal energy structure found by Demichelis et al. (2012).” But as we discussed in our work, the P6₃/mmc model cannot be used to index our electron diffraction data as well.

- Burgess & Bryce, *Solid State Nucl. Mag. Res.* 2015, 65: 75-83 (43Ca NMR, XRD): two structures (polytypes P3221 and C2) compatible with recorded 43Ca NMR spectra and both compatible with XRD; In this paper, the authors proposed two possible space groups of P3₂21 and C2. Unfortunately, none of them can be used to index our serial electron diffraction patterns.

- De La Pierre et al. 2014, *J. Phys Chem C* 2014, 118, 27493 (experimental and DFT Raman spectroscopy):two structures (polytypes P3221 and C2)have computed Raman spectrum that is compatible to experimental Raman spectra (referred to as multiple structure there)
Again, these proposed spaces groups cannot be used to index our serial electron diffraction patterns.

- Christy, *Cryst. Growth Des.* 2017, 17, 6, 3567–3578: reexamination of the whole literature, including the above references, and conclusions that it's a case of multiple structures/polytypism, with further exclusion of some theoretical structures based on crystallography.
This paper reviewed the advances in researchers’ understanding towards the structural features of vaterite. Although some researchers propose the concept of polytypic features, there is no direct imaging evidence until now. Also, there is no deep descriptions of this polytypic features.

After Christy's analysis, there is no longer a debate on the fact that vaterite exhibits polytypism/multiple structure. While it is true that some sort of doubt still exist about the actual space groups that can describe vaterite, and more images/evidence about its polytypism are welcome, many of the "unknown" features claimed in this paper have already been addressed.

It is true that the "unknown" features claimed in this paper have already been found in the past century since the vaterite was discovered. But this unknown feature was never well understood until now although many researchers realized its existence. We do not think the “unknown” features were addressed since there is no even direct imaging evidence and no detailed descriptions of this. For the first time, we not only directly reveal the polytypic features at the atomic scale resolution but also rationalize the occurrence of this polytypic features.

While there is value in having further evidence that supports vaterite polytypism, this paper does not match the novelty requirement of Nature Communication, and it does not contain any important advance of significance to specialists within the fields related to vaterite (mostly geochemistry, biology, chemistry). The authors should consider submitting this paper for consideration to a more specialised journal (e.g. J. Cryst. Growth) after reworking introduction and discussion through adding more details about the actual findings in the aforementioned literature.

Response: We cannot agree the reviewer on this. As indicated in above, researchers are still debating on the structures of vaterite. As stated by other researchers (Chemical Geology, 2014, 374: 84-91.), “Significant uncertainties thus exist on the disordered structure of vaterite, and no model can be considered as definitely accepted”. Uncovering the structural mystery of vaterite is very critical and fundamentally important for geochemistry, biology, chemistry as well. For geochemistry, the exact model would provide a more fundamental insights into the behavior of fluorine in biogenic vaterite (Geochimica et Cosmochimica Acta, 2021, 308: 384-392.); For biology, the exact model of vaterite would provide more enrich understanding of biophysics/ biochemistry (asteriscus), which play an important role in the teleost fish balance system (Feng Q. Principles of calcium-based biomineralization[M]//Molecular Biomineralization: Aquatic Organisms Forming Extraordinary Materials. Berlin, Heidelberg: Springer Berlin Heidelberg, 2011: 141-197, Morris R W, Kittleman L R. Piezoelectric property of otoliths[J]. Science, 1967, 158(3799): 368-370); For chemistry, knowing the exact structure of vaterite is a great help to design and synthesize the expected vaterite (Nature communications, 2017, 8(1): 15066, Advanced Materials, 2022: 2205088 and Crystal Growth & Design, 2018, 18(1): 331-337). The exact structure of vaterite has lots more to offer to climate change (Geochimica et Cosmochimica Acta, 2018, 236: 351-360) and biophotonic applications (Advanced Materials, 2021, 33(25): 2008484, Chemical Physics Letters, 435(1-3), 59-64). By means of systematic TEM characterizations, elaborate crystallographic analysis and machine learning aided molecular dynamics simulations with *ab initio* accuracy, we clarify the broadly discussed unknown features and reveal that vaterite can be regarded as a polytypic structure. The basic phase is a monoclinic lattice possessing pseudo-hexagonal symmetry. Direct imaging and atomic-scale simulations provide evidence that a single grain of vaterite can have three orientation variants. Additionally, we find that vaterite undergoes a second-order phase transition. To sum up, we believe that this paper realizes a complete atomic scale understanding of the structure of vaterite and has a great inspiration to the similar vaterite-structure and other vaterite- communities.

Reviewer #4 (Remarks to the Author):

A valuable contribution to understanding the complex real structure of vaterite, which is increasingly being recognized as an important biomineral. I have made some comments and corrections in notes on a pdf of the manuscript, which is attached.

Response: Thank the reviewer for the positive comments on our work. The changes of language description proposed by the reviewer have been fixed and highlighted in red color in the updated version.

Reviewers' Comments:

Reviewer #1:

Remarks to the Author:

Dear Authors,

Thank you very much for having made the effort, more detailing the effect of your models on "usual" XRD diagrams. It helped me much in "seeing" what the model brings:

1- Indeed, you are observing the "usual" extra peaks on XRD diagrams. These are located around 30° and 40° in 2θ for their main signatures.

2- The signature around 40° is roughly accounted for, this is why your GoF is decreased compared to the situation with no faults introduced. But the one at 30° not at all (on the diagrams you provided). I think we can discuss this differently as it is in your actual paper, for the following reasons.

I think you will agree with me that including faults using supercells adds more peaks (eventually broadened using micro strains) that help fitting some parts of the diagrams. The more faults you introduce, the more peaks you fit.

But if the faults are not corresponding to periodicities along the corresponding peaks, it does not help (case for the contribution at 30°).

Faults introduce narrow peaks, but requires lots of faulting to represent the whole extra signatures with both broad and narrow peaks.

Both signatures at 30 and 40° are composed of many diffraction lines, understandable when you treat the system as an incommensurately modulated structure (Steciuk work), and such an approach not only reproduces all the characteristics of the diagrams (all extra peaks), but assigns lines in 4D space, i.e. identifies modulations of the reciprocal space. All narrow and broad extra signatures are taken.

There is an intrinsic signal that does not favor the use of faults as the main responsible for extra signatures: they are always appearing, in all vaterites, synthetic or natural, with always the same intensities. How would faulting be the same in a pearl and in a synthetic powder, for instance, while crystal growing in such systems is very different ?

This is to help discussing the point: if a model "unlocks a mystery" it has to be better than the previously proposed ones. For vaterite up to now, the only model able to identify all signatures is the one proposed by Steciuk. BUT, your approach gives interesting lines. I would then propose you to modify your wording in title and text in order to let it describe the reality of the findings.

Because your model is clearly better the other ones already proposed by using faults, even if still lacking completeness in representing all peaks :-). Cause for sure some faulting exists in vaterite meriting some description.

The energetic calculation approach is also interesting, but has to be criticized. You tell in your answers: 41 meV/f.u. is larger than the gap between vaterite and laterite. Then I understand why the "0" variant cannot be here, it would have corresponded to a too strong "shaking" of the structure. How are the other variants comparing to the other gaps (calcite-aragonite-vaterite) ?

Again thank you very much for the effort in clarifying my mind.

sincerely

Daniel Chateigner

Reviewer #2:

Remarks to the Author:

All my comments were addressed. I appreciate the newly supplied XRD data and discussion about the scattering methods. They have also provided evidence of good enough lab-XRD data for the Rietveld analysis. I recommend this paper for publication, after adding XRD measurements details into Exp. Section.

Reviewer #3:

Remarks to the Author:

I do not question the quality of the experiments and the fact that this paper provides further evidence that can help understanding the structure of Vaterite. However, given that many of the conclusions strengthen evidence that has already been provided by others, this paper lacks the level of novelty required for Nature Comms and it would best fit into a more specialised journal

From the conclusions:

One of the main finding is that vaterite has either a C2 or C2/c structure.

Main literature presenting evidence in line with this:

- Mugnaioli et al, Angew. Chem. Int. Ed. 2012, 51, 7041 –7045; proposes C2/c and C-1 based on AED. (Note that C2 and C2/c are very similar, there's a loss of symmetry in C2 caused by a minor rotations of CO3 groups - experimentally this might not be able to be captured)
- Burgess & Bryce, Solid State Nucl. Mag. Res. 2015, 65: 75-83 (43Ca NMR, XRD) and De La Pierre et al. 2014, J. Phys Chem C 2014, 118, 27493 (experimental and DFT Raman spectroscopy): two amongst the many proposed structures (polytypes P3221 and C2) compatible with recorded spectra, but the techniques unable to distinguish whether one or the other (or a mix of both)
- Demichelis et al. Cryst. Growth. Des. 2013, 13, 2247 (DFT): two most stable structures C2 and P3221; C2/c is a transition state
- Christy, Cryst. Growth Des. 2017, 17, 6, 3567–3578 provides a crystallographic reason as to why C2/c isn't possible and C2 is.

Also:

"Vaterite has a pseudo-hexagonal feature with a pseudo sixfold axis along the [103]m direction, and three variants that are formed due to a 60° rotation along this direction."

This is also not a conclusion as it's been known for years (see aforementioned literature, and literature shared in my previous review)

The following is discussed here: Demichelis et al. Cryst. Growth. Des. 2013, 13, 2247 (DFT)

"The difference between the C2 and C2/c structures is characterized by the slightly different orientations of the carbonate groups within a single layer, which are sensitive to temperature, leading to different symmetries in vaterite."

Additionally, some of the papers above that have been cited have not been read accurately.

E.g.

Line 48-49 "orthorhombic structure (33, 34), monoclinic lattice and triclinic lattice (29, 30)."
- A. Le Bail, S. Ouhenia, D. Chateigner, Microtwinning hypothesis for a more ordered vaterite model. Powder Diffr. 26, 16-21 (2012).

This is orthorhombic - correct

34 - R. Demichelis, P. Raiteri, J. D. Gale, R. Dovesi, A new structural model for disorder in vaterite from first-principles calculations. CrystEngComm 14, 44-47 (2012). This excludes orthorhombic and analyses the hexagonal pool of structures, with P3221 as the most stable - so this would be pseudo-hexagonal

29 - R. Demichelis, P. Raiteri, J. D. Gale, R. Dovesi, The multiple structures of vaterite. Cryst. Growth. Des. 13, 2247-2251 (2013).

This analyses triclinic, monoclinic and hexagonal pools, proposing C2 and P3221 as the most stable structures - so this would be monoclinic and/or hexagonal

30- E. Mugnaioli et al., Ab initio structure determination of vaterite by automated electron diffraction. Angew. Chem. Int. Ed. 51, 7041-7045 (2012).

This proposes monoclinic C2/c or triclinic C-1

Line 125-126:

"These lattice parameters could match any of three monoclinic lattices including the C2, Cc, and C2/c structures proposed in previous work (29)"

Ref 29 excludes C2/c as it's a transition state. Crystallographic reasons for this is discussed in

Christy, Cryst. Growth Des. 2017, 17, 6, 3567–3578

Response to reviewers' comments

Reviewer #1 (Remarks to the Author):

Dear Authors,

Thank you very much for having made the effort, more detailing the effect of your models on "usual" XRD diagrams. It helped me much in "seeing" what the model brings:

Response: We thank the reviewer for spending the time reviewing our work and providing us the invaluable suggestions to enhance the quality of our work. We have addressed your following comments one by one in the following texts.

1- Indeed, you are observing the "usual" extra peaks on XRD diagrams. These are located around 30° and 40° in 2θ for their main signatures.

2- The signature around 40° is roughly accounted for, this is why your GoF is decreased compared to the situation with no faults introduced. But the one at 30° not at all (on the diagrams you provided). I think we can discuss this differently as it is in your actual paper, for the following reasons.

I think you will agree with me that including faults using supercells adds more peaks (eventually broadened using micro strains) that help fitting some parts of the diagrams. The more faults you introduce, the more peaks you fit.

But if the faults are not corresponding to periodicities along the corresponding peaks, it does not help (case for the contribution at 30°).

Faults introduce narrow peaks, but requires lots of faulting to represent the whole extra signatures with both broad and narrow peaks.

Both signatures at 30 and 40° are composed of many diffraction lines, understandable when you treat the system as an incommensurately modulated structure (Steciuk work), and such an approach not only reproduces all the characteristics of the diagrams (all extra peaks), but assigns lines in 4D space, i.e. identifies modulations of the reciprocal space. All narrow and broad extra signatures are taken.

There is an intrinsic signal that does not favor the use of faults as the main responsible for extra signatures: they are always appearing, in all vaterites, synthetic or natural, with always the same intensities. How would faulting be the same in a pearl and in a synthetic powder, for instance, while crystal growing in such systems is very different?

Response: Yes, we strongly agree with you that including stacking disorders within supercells adds more peaks that help fitting the diagrams. The more faults are introduced, the more peaks are fitted. We believe

the heavily faulted features can roughly account for the signatures peak at around 30° and 40° , which are composed of many diffraction lines. As shown in Fig. X1, an XRD simulation of the C2 structure, the $(222)_m$ and $(004)_m$ peaks at located around 40.6° and 42.9° respectively, which are labeled in Fig. X2. Similarly, the $(220)_m$ and $(11-3)_m$ at around 30° are also labeled in Fig. X2. Based on the diffraction data shown in Fig. X2, it is predictable that by introducing many stacking faults within C2 structure the peaks around 30° and 40° will be broadened due to the evident streaking features along $(00l)$ planes, where l is an integer. Steciuk *et al.* tried to resolve the structure of vaterite using the 4D spaces method in a high dimensional space, which is also a great achievement in interpreting the supercell features of vaterite. Based on our work, the polytypic features in vaterite are on a very small scale. The localized stacking order/disorder cannot be the same among different areas even in single grain interior. Therefore, it is almost impossible for the pearl and synthetic vaterite powder to share the same stacking configurations although all of them have polytypic features. By the way, as shown in Fig. X3, the synthetic vaterite powders also have polytypic features.

Fig. X1. The simulated XRD patterns of vaterite with C2 structure.

Fig. X2. Electron diffraction pattern along $[1-10]$ direction of C2 structure.

Fig. X3. Stacking disorder within synthetic vaterite. (a) SEM image of the synthetic vaterite. (b) HRTEM image of the synthetic vaterite. (c) Atomic resolution HAADF image along the $[010]_m$ zone axis again showing the polytypic features within synthetic vaterite. (d) Strain map (ϵ_{xy}) of the image shown in (c) obtained by geometric phase analysis (GPA).

This is to help discussing the point: if a model "unlocks a mystery" it has to be better than the previously proposed ones. For vaterite up to now, the only model able to identify all signatures is the one proposed by Steciuk. BUT, your approach gives interesting lines. I would then propose you to modify your wording in title and text in order to let it describe the reality of the findings. Because your model is clearly better the other ones already proposed by using faults, even if still lacking completeness in representing all peaks :-). Cause for sure some faulting exists in vaterite meriting some description.

Response: Thank you for the suggestion. We have modified our title to “Unlocking the mysterious polytypic features within vaterite CaCO_3 ” to reflect your comments. The reason that Steciuk’s 4D spaces method gives better completeness is due to the method itself is a reverse method. The 4D spaces method is proposed and constructed based on all the peaks in diffraction data. Therefore, this method can index all the peaks in the XRD. In our work, we are trying to reveal the structures through a direct way with the model proposed based on atomic scale images. Thus, the simulations based on our supercell model can infinitely approach the experimental peaks but cannot achieve perfect match since we can never build a supercell model having the same stacking order/disorder as the real materials with micrometer scale.

The energetic calculation approach is also interesting, but has to be criticized. You tell in your answers: 41 meV/f.u. is larger than the gap between vaterite and laterite. Then I understand why the "0" variant cannot be here, it would have corresponded to a too strong s"shaking" of the structure. How are the other variants comparing to the other gaps (calcite-aragonite-vaterite) ?

Response: In our calculations, the energies of the other two variants (“+” and “-”) are nearly identical (less than the root means square error of the potential). Therefore, we believe that both variants will occur in vaterite, which is also consistent with the results of our calculations and experiments. Among the three crystal structures of calcium carbonate, the energy gap between calcite and vaterite exhibits the most significant. We have employed this value as a benchmark to compare the energy of different variants, primarily to illustrate the instability of the "0" arrangement.

Again thank you very much for the effort in clarifying my mind.
sincerely
Daniel Chateigner

Reviewer #2 (Remarks to the Author):

All my comments were addressed. I appreciate the newly supplied XRD data and discussion about the scattering methods. They have also provided evidence of good enough lab-XRD data for the Rietveld analysis. I recommend this paper for publication, after adding XRD measurements details into Exp. Section.

Response: We thank the reviewer for spending the time reviewing our work and providing us with many constructive comments. We have added the following XRD measurements details in the Method session.

“Powder diffraction data were collected at room temperature on a STOE-STADI-P powder diffractometer equipped with an asymmetric curved Germanium monochromator (Cu $K_{\alpha 1}$ radiation, $\lambda = 1.54056 \text{ \AA}$) and one-dimensional silicon strip detector (MYTHEN2 1K from DECTRIS). The line focused Cu X-ray tube was operated at 40 kV and 40 mA. Powder was packed in an 8 mm metallic mask and sandwiched between two polyimide layers of tape. Sample holder was span during collection. Intensity data from 10 to 80 degrees two theta were collected over a period of 12.5 hours with an overall 0.005 degrees step. The instrument was calibrated against a NIST Silicon standard (640d) prior to the measurement.”

Reviewer #3 (Remarks to the Author):

I do not question the quality of the experiments and the fact that this paper provides further evidence that can help understanding the structure of Vaterite. However, given that many of the conclusions strengthen evidence that has already been provided by others, this paper lacks the level of novelty required for Nature Comms and it would best fit into a more specialized journal.

Response: We thank the reviewer for spending the time reviewing our work and acknowledging the quality of our study. However, we do not agree with the reviewer's comment that it is lacking novelty. The novelty of our work is listed as follows:

- 1> As stated in the introduction and Supplementary Note I, structure of vaterite has been debated for about a century. Until now, there is even a single atomic resolution image, which can directly reveal the structures of vaterite. In almost all the available literature, researchers are trying to resolve the structural features of vaterite by using the bulk measurements method such as XRD, neutron diffraction and Raman. The bulk measurements methods are very limited since the nanoscale inhomogeneous features are always averaged out. Indeed, some researchers are trying to resolve the structure of vaterite using electron based diffraction methods, which have a higher spatial resolution than the X-ray and neutron based diffraction methods. But this is still very limited without complementary direct atom resolved imaging information in real space. In other aspects, although high resolution TEM (HRTEM) was also used to reveal the structural features of vaterite, it is also very limited since HRTEM only provides the phase contrast image which always introduces the delocalization. Considering that atom columns in HRTEM image cannot be directly related to atom locations, it is very limited for resolving the features of heavily faulted complicate structures. Atomic resolution high angle annular dark field (HAADF) imaging is a perfect tool to resolve the features of vaterite. Although many researchers are aware of this, it is still very challenging to obtain the high-quality atomic resolution HAADF images of vaterite because of the poor crystallinity. By overcoming many technical challenges, for the first time, our group directly reveals the polytypic features within vaterite through the high-quality atomic scale resolution images.
- 2> The polytypic features extracted from atomic resolution images are consistent with the serial electron diffraction data and bulk XRD measurements. Additionally, we developed the theoretical models by using the machine learning aided MD simulations molecular dynamics simulations with *ab initio* accuracy. By directly growing the vaterite from amorphous liquid, we obtain a polytypic structure theoretically, which agrees well with our experimental observations.
- 3> By performing systemic experimental work on vaterites from different sources, we find that the polytypic features are very general, which is independent of the originations although the environment may slightly tailor the local stacking of vaterite.

From the conclusions:

One of the main finding is that vaterite has either a C2 or C2/c structure.

Response: This is only one fact of our study. But we go far beyond that. Based on theoretical calculations, we find that C2 (low symmetry phase) is more stable at low temperatures and C2/c (high symmetry phase) is more stable at high temperatures. For the first time, we uncover that the transformation between the low- and high-symmetry phases can be well described by a second-order phase transition with a critical point of approximately 190 K. Additionally, we find that vaterite with basic monoclinic lattice has a pseudohexagonal feature with a pseudo sixfold axis along the $[103]_m$ direction. There are three orientation variants that are formed due to a 60° rotation along $[103]_m$ in single vaterite grain. We also reveal that the stacking sequence “0” is absent, while “+” and “-” sequences are randomly distributed along the $[103]_m$ direction. Regarding the other comments related to some specific literatures, we have addressed them one by one in the following.

Main literature presenting evidence in line with this:

- Mugnaioli et al, *Angew. Chem. Int. Ed.* 2012, 51, 7041 –7045; proposes C2/c and C-1 based on AED. (Note that C2 and C2/c are very similar, there’s a loss of symmetry in C2 caused by a minor rotations of CO3 groups - experimentally this might not be able to be captured)

Response: In this paper, the authors tried to use automated electron diffraction technique to resolve the structure mystery of vaterite and proposed that vaterite likely has the C2/c and C1 structure. As we stated previously, although electron diffraction technique has a high spatial resolution (tens of nm scale), it is still not enough and very limited in resolving the complex structures with heavily faulted features. The complementary atom resolved images with sub angstrom spatial resolution are still very demanding. In this paper, the authors provided the six-layer triclinic model and two-layer monoclinic model, but still emphasized that C2/c model showed relatively high residuals even for nanocrystalline samples. The heavily faulted features are even not mentioned and discussed in this paper.

- Burgess & Bryce, *Solid State Nucl. Mag. Res.* 2015, 65: 75-83 (43Ca NMR, XRD) and De La Pierre et al. 2014, *J. Phys Chem C* 2014, 118, 27493 (experimental and DFT Raman spectroscopy): two amongst the many proposed structures (polytypes P3221 and C2) compatible with recorded spectra, but the techniques unable to distinguish whether one or the other (or a mix of both)

Response: As the authors (*Solid State Nucl. Mag. Res.* 2015, 65: 75-83) themselves said, some models including the DFT-predicted GGA, LDA, and P2₁2₁2₁ as well as the monoclinic C2/c and C1 structures are inconsistent with the PXRD data. The P3₂21 model (or its stereoisomeric P3₁21 structure) and the monoclinic C2 model provide the best simultaneous agreement between the experimental and calculated NMR data, and the experimental and simulated PXRD data. The authors also suggested that the data

reported herein would remain valuable for the cross-validation of further proposals for the true structure of vaterite.

In another study (J. Phys Chem C 2014, 118, 27493), the comparison indicates that both the two lowest energy structures C2 and P3₂21, are compatible with Raman measurements on vaterite. Simulated Raman spectra do not make it possible to discriminate as to whether the first, the second or both structures are present in the experimental samples. Again, as the authors said, to verify this hypothesis, further investigation is required in order to determine the activation barriers for the interconversion between all of the different energy minima.

- Demichelis et al. Cryst. Growth. Des. 2013, 13, 2247 (DFT): two most stable structures C2 and P3₂21; C2/c is a transition state.

Response: In this paper, the authors studied the energy landscape of many groups such as P3₂21(P3₁21), P112₁, P6₅22 (P6₁22), P6₅ (P6₁), C1, C-1, C2, C2/c and Cc, and found that C2 and P3₂21 were the most stable structures. But the key thing is that vaterite cannot be described by single lattice with any space groups and C2 is only a basic structure. The heavily faulted features and details of the polytypism are not mentioned and discussed in this study.

- Christy, Cryst. Growth Des. 2017, 17, 6, 3567–3578 provides a crystallographic reason as to why C2/c isn't possible and C2 is.

Response: In this paper, the authors discussed the impossible, possible but unlikely and likely structures. The authors proposed to use the “maximum-degree-of-order (MDO)” theory to discuss the intrinsic stabilities of vaterite structures with different space groups. Among many possible space groups, the authors indeed find that C2/c is less stable than C2 from the view of crystallographic reason. But there is no direct experimental evidence to prove this in this study. Again, the key thing for vaterite is not the C2 or C2/c since none of them can be used to explain the experimental XRD results. Additionally, the heavily faulted features, orientation domains and second phase transformation, which are representative features behind the monoclinic lattice, are not mentioned or even discussed in this study.

Also:

“Vaterite has a pseudo-hexagonal feature with a pseudo sixfold axis along the [103]_m direction, and three variants that are formed due to a 60° rotation along this direction.”

This is also not a conclusion as it's been known for years (see aforementioned literature, and literature shared in my previous review)

Response: In this study (Cryst. Growth Des. 2017, 17, 6, 3567–3578), the word “variant” occurs eighteen times. However, the meaning of the variant in this paper is totally different from the ones we discussed in our paper. In this study, the authors claimed “three desymmetrized variants of the polytype can be distinguished, which are here labeled 6H-P6₅, 6H-P3₂21, and 6H-P112₁”. The variants mentioned in Christy’s work refers to three different space groups with similar symmetries. However, the variant in our work means orientation variants which are generated from the continuous 60° rotation along the pseudo-6 fold axis. But these three orientation variants share the same space group of C2. In another paper (Powder Diffraction, 2011, 26(1): 16-21), the authors put forward a microtwinning hypothesis with three domains rotated by 120° along the orthorhombic an axis based on the space group of Ama2. This information was extracted based on the HRTEM images, which is inaccurate because the features does not meet the definition of the twin. To define the features of a twin, at least five twin elements (K1, h1, K2, h2, twin disconnection and shear angles) are needed. But none of them are even mentioned in this paper. Additionally, the authors found that five weak superstructure reflections seen in single-crystal and powder diffraction experiments still cannot be unexplained based on their microtwinning hypothesis.

The following is discussed here: Demichelis et al. Cryst. Growth. Des. 2013, 13, 2247 (DFT)
“The difference between the C2 and C2/c structures is characterized by the slightly different orientations of the carbonate groups within a single layer, which are sensitive to temperature, leading to different symmetries in vaterite.”

Response: Unfortunately, we don't find a similar discussion in this work (Cryst. Growth. Des. 2013, 13, 2247). The authors performed the DFT calculations at room temperature and do not consider the structural stability at high temperatures. However, in our work, we performed machine learning aided molecular dynamic simulations, which account the effects of temperatures. Theoretically, we directly grow the vaterite from a high temperature amorphous liquid state into a crystal structure with highly faulted features. Additionally, we uncover the second phase transformation in vaterite which is never discussed in any previous literatures.

Additionally, some of the papers above that have been cited have not been read accurately.

E.g.

Line 48-49 “orthorhombic structure (33, 34), monoclinic lattice and triclinic lattice (29, 30).”^[SEP]33 - A. Le Bail, S. Ouhenia, D. Chateigner, Microtwinning hypothesis for a more ordered vaterite model. Powder Diffr. 26, 16-21 (2012).

This is orthorhombic – correct.

34 - R. Demichelis, P. Raiteri, J. D. Gale, R. Dovesi, A new structural model for disorder in vaterite from first-principles calculations. CrystEngComm 14, 44-47 (2012).^[SEP]This excludes orthorhombic and analyses the hexagonal pool of structures, with P3221 as the most stable - so this would be pseudo-hexagonal

29 - R. Demichelis, P. Raiteri, J. D. Gale, R. Dovesi, The multiple structures of vaterite. Cryst. Growth. Des. 13, 2247-2251 (2013).

This analyses triclinic, monoclinic and hexagonal pools, proposing C2 and P3221 as the most stable structures - so this would be monoclinic and/or hexagonal

30- E. Mugnaioli et al., Ab initio structure determination of vaterite by automated electron diffraction. *Angew. Chem. Int. Ed.* 51, 7041-7045 (2012).

This proposes monoclinic C2/c or triclinic C-1

Response: Space P3₂21 is a rhombohedral lattice but not the psedu-hexagonal structure. We do not prefer to use “psedu-hexagonal” to represent a “rhombohedral lattice”. Our psedu-hexagonal feature is based on C2 structure which has a monoclinic lattice. This is totally different from the space group of P3₂21. We have changed our description slightly for the previously inaccurate citations in the updated version.

“such as the hexagonal or rhombohedral structures^{29,32,33}, orthorhombic structure³⁴, monoclinic lattice and triclinic lattice^{29,30}”

Line 125-126:

“These lattice parameters could match any of three monoclinic lattices including the C2, Cc, and C2/c structures proposed in previous work (29)”

Ref 29 excludes C2/c as it’s a transition state. Crystallographic reasons for this is discussed in Christy, *Cryst. Growth Des.* 2017, 17, 6, 3567–3578

Response: We already addressed this in the previous question. In this paper, the authors discussed the impossible, possible but unlikely and likely structures. The authors proposed to use the “maximum-degree-of-order (MDO)” theory to discuss the intrinsic stabilities of vaterite structures with different space groups. Among many possible space groups, the authors indeed find that C2/c is less stable the C2 from the view of crystallographic reason. But there is no direct experimental evidence to prove this in this study. Again, the key thing for vaterite is not the C2 or C2/c since neither of them can be used to explain the experimental XRD results. Additionally, the heavily faulted features, orientation domains and second phase transformation, which are representative features behind the monoclinic lattice, are not mentioned or even discussed in this study.